# Interpretation of T cell states from single-cell transcriptomics data using reference atlases

Massimo Andreatta [1,2], Jesus Corria-Osorio[1], Sören Müller [3], Rafael Cubas[4], George Coukos [1] & Santiago J. Carmona [1,2✉]

Single-cell RNA sequencing (scRNA-seq) has revealed an unprecedented degree of immune cell diversity. However, consistent definition of cell subtypes and cell states across studies and diseases remains a major challenge. Here we generate reference T cell atlases for cancer and viral infection by multi-study integration, and develop ProjecTILs, an algorithm for reference atlas projection. In contrast to other methods, ProjecTILs allows not only accurate embedding of new scRNA-seq data into a reference without altering its structure, but also characterizing previously unknown cell states that "deviate" from the reference. ProjecTILs accurately predicts the effects of cell perturbations and identifies gene programs that are altered in different conditions and tissues. A meta-analysis of tumor-infiltrating T cells from several cohorts reveals a strong conservation of T cell subtypes between human and mouse, providing a consistent basis to describe T cell heterogeneity across studies, diseases, and species.

[1] Department of Oncology, Lausanne Branch, Ludwig Institute for Cancer Research, CHUV and University of Lausanne, Lausanne, Epalinges, Switzerland. [2] Swiss Institute of Bioinformatics, Lausanne, Switzerland. [3] Department of Bioinformatics and Computational Biology, Genentech, South San Francisco, CA, USA. [4] Department of Translational Oncology, Genentech, South San Francisco, CA, USA. ✉email: santiago.carmona@unil.ch

In response to malignant cells and pathogens, mammals (and presumably most jawed vertebrates) mount an adaptive immune response characterized by a finely-tuned balance of several specialized T cell subtypes with distinct migratory and functional properties and metabolic lifestyles. Occasionally, however, malignant cells and pathogens escape immune control, leading to cancer and chronic infections. Antigen persistence in cancer and chronic infections profoundly alters T cell differentiation and function, leading antigen-specific cells into a collection of transcriptional and epigenetic states commonly referred to as "exhausted"[1]. The complexity and plasticity of T cells make the study of adaptive immune responses in these contexts particularly challenging.

In recent years, single-cell RNA-sequencing (scRNA-seq) enabled unbiased exploration of T cell diversity in health, disease and response to therapies at an unprecedented scale and resolution. While the presence of tumor-infiltrating T lymphocytes (TILs) in cancer lesions has been broadly associated with improved prognosis and response to immune checkpoint blockade[2,3], scRNA-seq has uncovered a great diversity within TILs, suggesting that distinct TIL states contribute differently to tumor control and response to immunotherapies[4–6]. However, a comprehensive definition of T cell "reference" subtypes remains elusive. Poor resolution of T cell heterogeneity remains a limiting factor towards understanding the effect of induced perturbations, such as therapeutic checkpoint blockade and genetic editing, in particular when these simultaneously affect the frequencies and intrinsic features of T cell subtypes.

A major challenge towards the construction of reference single-cell atlases is the integration of gene expression datasets produced from heterogeneous samples from multiple individuals, tissues, batches and generated using different protocols and technologies. Several computational methods have been developed to correct technical biases introduced by handling experiments in batches, and to align datasets over their biological similarities[7–13]. We previously developed STACAS[14], a bioinformatic tool for scRNA-seq data integration specifically designed for the challenges of integrating heterogeneous datasets characterized by limited overlap of cell subtypes. This is particularly relevant for the construction of T cell atlases, where differences between datasets are not merely the result of technical variation of handling samples in different batches, but rather due to subtypes of highly variable frequency, and in many cases subtypes that are entirely missing from one or more samples as a result of study design or biological context. While whole-organism single-cell atlases are very powerful to describe the global properties of cell populations[15], only by constructing specialized atlases for individual cell types can one achieve the level of resolution required to discriminate the spectrum of transcriptional states that can be assumed by each cell type.

A second outstanding challenge in single-cell data science is the mapping of single cells to a reference atlas[16], and several methods have been proposed to address this task[17–20]. These methods allow mapping cluster annotations of a reference atlas to individual cells of a query dataset (also referred to as "label transfer"). However, these methods define a new embedding space specific for the query, not preserving the integrity of the reference atlas space upon mapping. The ability to embed new data points into a stable reference map would enable robust, reproducible interpretation of new experiments in the context of curated and annotated cell subtypes and states. In addition, different conditions (e.g. pre- vs. post-treatment, or mutant vs. wild-type) could be compared over a unified transcriptomic reference landscape. In the absence of reliable reference cell atlases—and computational tools to project new data onto these atlases—researchers must rely on unsupervised, manual annotation of their data, a time-consuming and, to a certain degree, subjective process.

In this work, we develop ProjecTILs, a computational framework for the projection of new scRNA-seq data into reference atlases. In contrast to other methods, ProjecTILs enables mapping of new data into a reference atlas without altering the reference space, as well as detecting and characterizing previously unknown cell states that "deviate" from the reference subtypes. We demonstrate the robustness of the method by interpreting the effects of T cell perturbations in multiple model systems of cancer and infection, enabling the comparison of cell states from multiple studies over a stable system of coordinates. Finally, ProjecTILs analysis of T cell heterogeneity and clonal structure across patients, tissues, and cancer types, shows a high degree of conservation between human and mouse TIL states and provides insights into the differentiation of CD8$^+$ T cells in cancer.

## Results

**A cross-study reference atlas of tumor-infiltrating T cell states.** With the goal to construct a comprehensive reference atlas of T cell states in murine tumors, we collected publicly available scRNA-seq data from 21 melanoma and colon adenocarcinoma tumors (see "Methods"). In addition, we generated scRNA-seq data from four tumor-draining lymph node samples (MC38_dLN dataset, see "Methods"). After data quality checks and filtering pure αβ T cells, our database comprised expression profiles of 16,803 high-quality single-cell transcriptomes from 25 samples from six different studies (Supplementary Table 1).

Substantial batch effects are typically observed between single-cell experiments performed on different days, by different labs, or using different single-cell technologies. Without batch-effect correction, cells tend to cluster by study rather than by cell type (Fig. 1a). We applied the STACAS algorithm[14] to integrate the datasets over shared cell subtypes and combine them into a unified map (Fig. 1b, Supplementary Fig. 1). Unsupervised clustering and gene enrichment analysis, supported by T cell supervised classification by TILPRED[21] (Fig. 1c), allowed annotating areas of the reference map into "functional clusters" (Fig. 1d), characterized by known gene expression signatures of specific T cell subtypes. We observed a distinct separation between CD4$^+$ and CD8$^+$ T cells, which could be further divided into subgroups. In particular, we identified a cluster of naive-like CD8$^+$ T cells (which may consist of naive as well as central memory cells) and a smaller cluster of naive-like CD4$^+$ T cells, co-expressing *Tcf7* and *Ccr7* while lacking cytotoxic molecules and activation features such as *Pdcd1* and *Tnfrsf9*/4-1BB; a cluster of effector-memory (also abbreviated as EM) CD8$^+$ T cells, co-expressing *Tcf7* and granzymes (most prominently *Gzmk*), with low to intermediate expression of *Pdcd1*; an "early-activation" state of CD8$^+$ T cells, with an intermediate profile between the naive-like and the EM CD8$^+$ types; a CD8$^+$ terminally-exhausted (Tex) effector cluster, characterized by high expression of granzymes, multiple inhibitory receptors (*Pdcd1*, *Ctla4*, *Lag3*, *Tigit*, *Havcr2*/TIM-3, etc.) and *Tox*[21,22]; a CD8$^+$ precursor-exhausted (Tpex) cluster, with co-expression of *Tcf7*, *Pdcd1*, *Ctla4*, *Tox* but low expression of *Havcr2* or granzymes[21,23,24]; a cluster of CD4$^+$ Th1-like cells, expressing IFN-gamma receptor 1 (*Ifngr1*) and *Fasl*[25]; a CD4$^+$ follicular-helper (Tfh) population[25,26], with a pronounced expression level of *Cxcr5*, *Tox,* and *Slamf6*; and a cluster of regulatory T cells (Treg), identified by *Foxp3* (Fig. 1d–f). As expected, while TIL samples were mostly enriched in Tex, Tpex, and Treg subtypes, tumor-draining lymph nodes were enriched in naive-like and follicular helper cells (Fig. 1g). We confirmed that clusters identified by unsupervised analysis of individual datasets were largely consistent with corresponding "functional clusters" shared by multiple datasets in the integrated reference atlas (Supplementary Fig. 1).

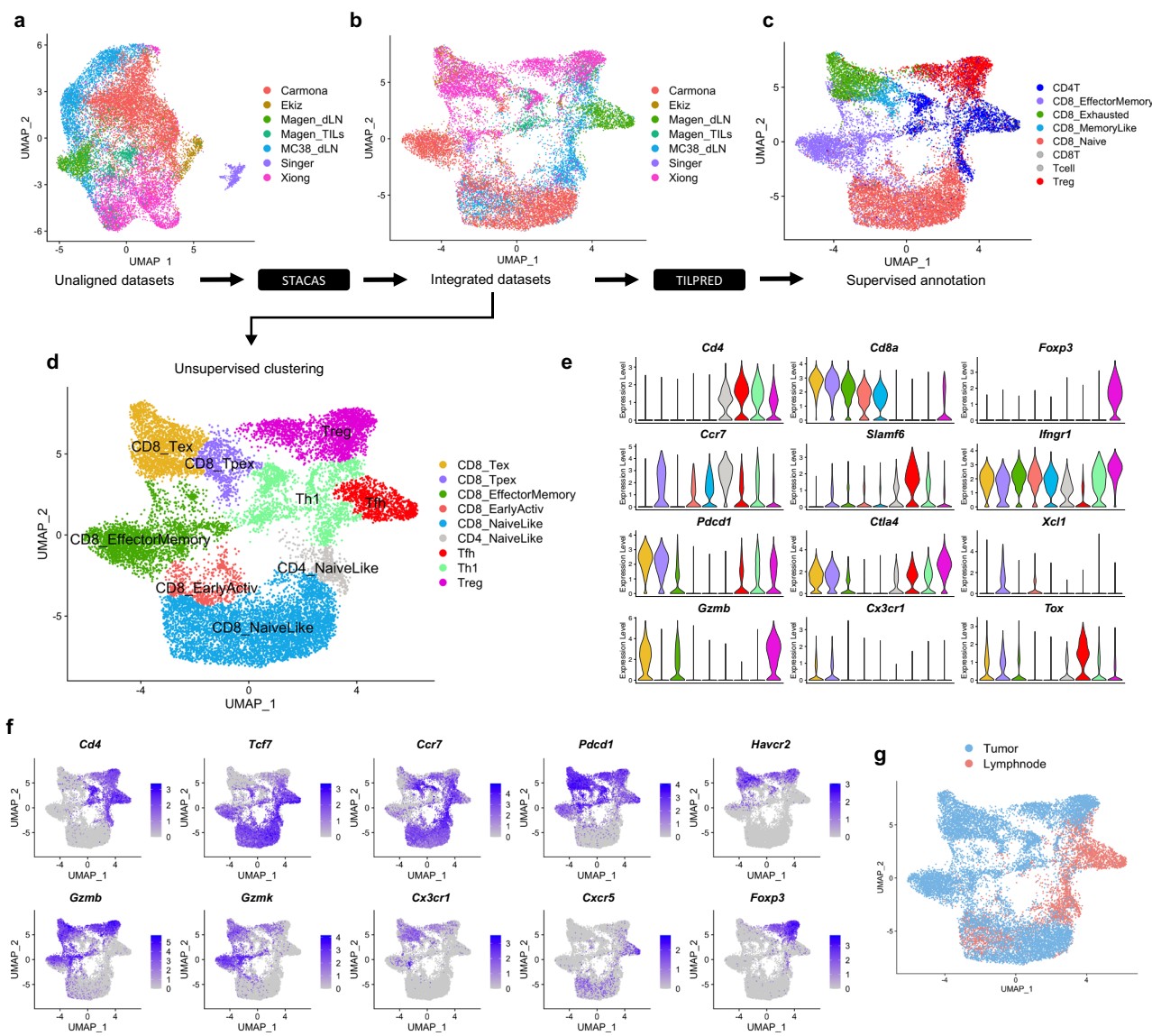

**Fig. 1 Building a reference map of TIL transcriptomic states. a** Uniform Manifold Approximation and Projection (UMAP) plots of single-cell transcriptomic profiles from different studies, before batch-effect correction (i.e., unaligned datasets); (**b**) Same plot for integrated datasets after STACAS alignment: successful dataset integration mitigates batch effects while preserving biological differences between T cell subtypes; (**c**) Supervised T cell subtype classification by TILPRED shows that, after alignment, cells cluster mainly by cell subtype rather than by dataset of origin; (**d**) Unsupervised clusters were annotated as nine functional states based on TILPRED prediction, as well as by (**e**) average expression of marker genes in each cluster and by (**f**) single-cell expression of key marker genes over the UMAP representation of the map; (**g**) Reference atlas colored by tissue of origin (tumor and draining lymph node). An interactive reference TIL atlas can be explored online at http://tilatlas.unil.ch.

Overall this reference atlas summarizes TIL diversity using nine broad cell subtypes with distinct phenotypes, functions, metabolic lifestyles, and preferential tissue distributions, and strongly supported by experimental evidence in murine models. An interactive interface of the TIL atlas, allowing the exploration of T cell subtypes and gene expression over the reference map can be accessed at http://TILatlas.unil.ch.

**Accurate projection of scRNA-seq data onto reference atlases**. In order to enable interpretation of new datasets in the context of reference T cell subtypes, we developed ProjecTILs, a computational method for the projection of scRNA-seq data onto a reference atlas. The essential input to ProjecTILs is the single-cell expression matrix of the query dataset, e.g., in UMI counts or TPMs, where rows represent the genes and columns represent the individual cells. The pre-processing steps (Fig. 2a) normalize

scRNA-seq data using a log-transformation (if provided non-normalized) and filter out non-T cells (see "Methods"). In order to reduce batch effects between the query and the reference map, the STACAS/Seurat integration procedure[14] is used to align the query to the reference, and in this way correct the expression matrix of the query dataset (Fig. 2b, see "Methods"). The corrected query matrix can then be projected onto reduced-dimensionality representations (e.g., PCA, UMAP) of the reference, effectively bringing them into the same reference space. To this end, the algorithm computes the PCA rotation matrix of the reference map, which contains the coefficients to linearly transform gene expression into PCA loadings (i.e., the eigenvectors and their relative eigenvalues); the same PCA rotation matrix is then also applied to the query set (Fig. 2c). Likewise, the UMAP transformation (allowing the computation of UMAP coordinates from PCA loadings) is applied to the

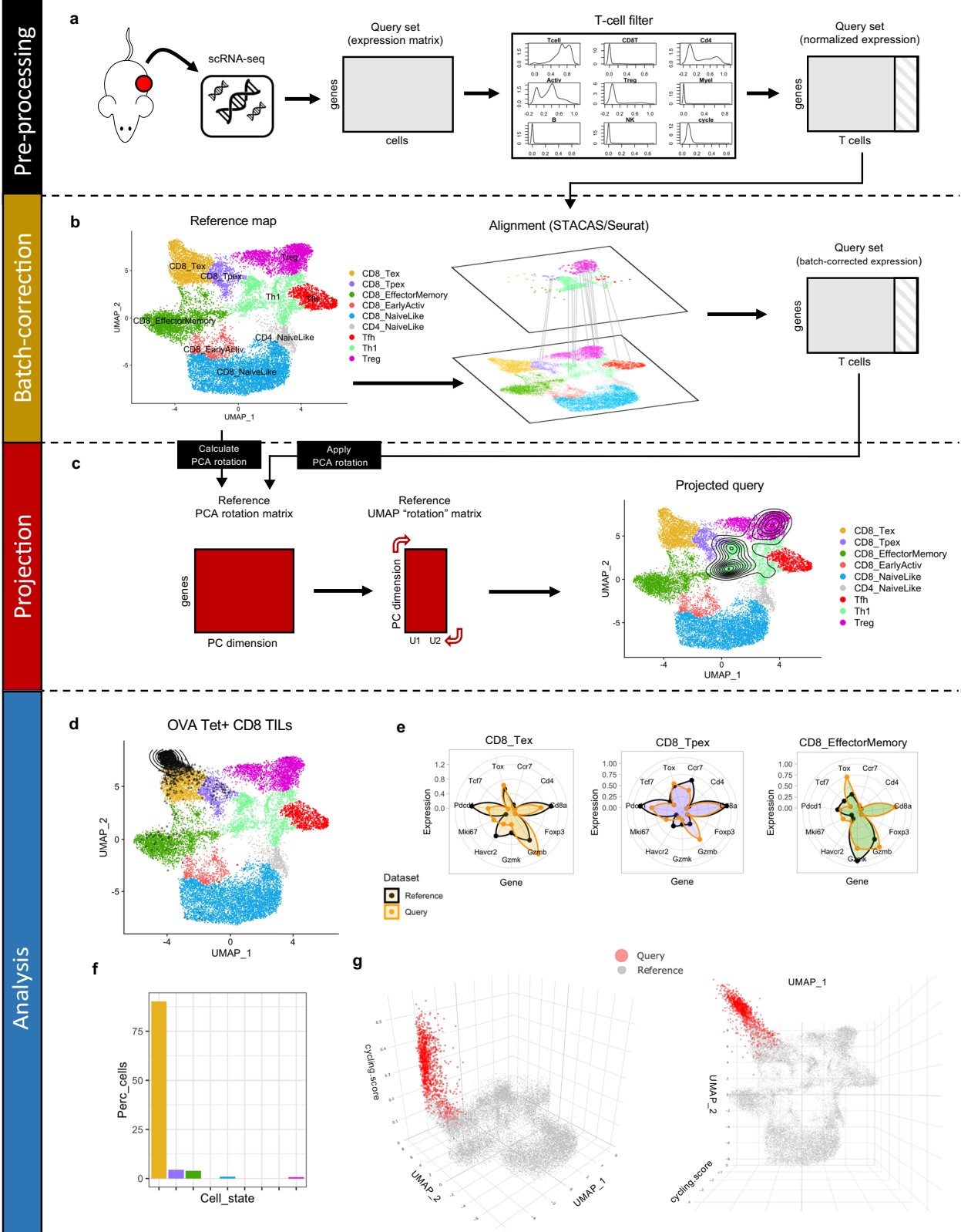

query set to project it into the original UMAP embedding of the reference map. Note that while the expression counts of the query set and their embedding into reduced spaces are modified by the alignment procedure, those of the reference map are not, and its low-dimensional visualizations remain unaltered; different query sets, or experiments containing different conditions can therefore be compared over the same reference map. After

projection, a nearest-neighbor classifier predicts the subtype of each query cell by a majority vote of its annotated nearest neighbors (either in PCA or UMAP space) in the reference map. Benchmarking ProjecTILs by cross-validation experiments showed a high accuracy (>90%) both for the projection (Supplementary Fig. 2, and "Methods") and classification tasks, significantly outperforming Azimuth/Seurat 4[27] and scmap[17],

**Fig. 2 The ProjecTILs analysis workflow. a** The essential input to ProjecTILs is a query dataset in the form of a gene expression matrix. Pre-processing steps include data normalization and filtering of non-T cells. **b** The normalized, filtered gene expression matrix is aligned to the reference map using STACAS, to bring the query data into the same scale as the reference map. **c** The PCA rotation matrix and UMAP transformation calculated on the reference map are applied to the query set, effectively embedding it into the same space of the reference map, and allowing their direct comparison and joint visualization. **d–g** Projection of tumor-specific tetramer+ CD8+ TIL single-cell data from Miller et al. **d** Predicted coordinates of the projected query in UMAP space as density contours. **e** Gene expression signature of query cells (orange) and reference cells (black) for the three most represented T cell subtypes; average gene expression for the reference is normalized between 0 and 1. **f** Percentage of cells predicted by the algorithm for the nine cell states of the reference atlas; over 90% of total cells are predicted to be CD8+ terminally exhausted cells (CD8_Tex). **g** UMAP plot augmented with cell cycling score on the z axis (side and top view); CD8_Tex cells for the query dataset are shown in red.

two alternative methods for reference-based single-cell data analysis (Supplementary Fig. 3).

As an illustrative application of ProjecTILs to analyze a query scRNA-seq dataset, we projected onto the reference murine TIL map a dataset of tumor-specific CD8+ T cells isolated by tetramer staining from untreated B16 melanoma tumors expressing chicken ovalbumin (OVA), from the study by Miller et al.[28]. Consistently, ProjecTILs assigned the great majority (90.2%) of tumor-specific cells to the Tex subtype, while small fractions were assigned to the Tpex (4.4%) and EM-like (3.9%) compartments (Fig. 2d–f). The expression profile of CD8_Tex cells matches well with the reference map for a panel of marker genes, with a pronounced overexpression of *Gzmb* and the proliferation marker *Mki67* (Fig. 2e), as expected by antigen-induced activation[29,30]. ProjecTILs allows visualizing an additional dimension on the z axis together with the 2D UMAP representation. In this case, we chose to plot a cell cycling signature score, which reveals a striking proliferative signal for the cells in the query dataset (Fig. 2g). Taken together, these results are in agreement with experimental observations and well compatible with the notion that tumor-specific CD8+ T cells, as a result of continued antigenic stimulation in the tumor, become enriched in a highly expanded, terminally exhausted state[28,31].

**ProjecTILs reveals altered gene programs following T cell perturbations.** CD8+ T cells depend on the microRNA 155 (miR-155) to acquire anti-tumoral or anti-viral effector functions[32,33]. To interpret the impact of miR-155 deficiency on the TIL landscape, we submitted to ProjecTILs the data from Ekiz et al.[34], which consist of total immune (CD45+) single cells from untreated B16 melanoma tumors growing in miR-155 T cell conditional knock-out (KO) mice, as well as in wild type (WT) mice. Note that, although the WT sample from this study was included in the construction of the reference map, for this analysis it was re-projected using the same pipeline applied to the miR-155 KO sample. This ensures that the two conditions were processed and projected uniformly and that they could be compared over the same reference map. Projecting these data onto the reference TIL atlas revealed that in WT mice, the majority of TILs correspond to CD8_Tex, with smaller populations of other cell types. Conversely, CD8+ TILs from miR-155 KO mice were mostly projected to the naive-like compartment (Fig. 3a). Moreover, while T cells accounted for 16% of the total immune infiltrate in WT mice, they were reduced to 8% in KO mice. Consistently with the predicted change in the dominant T cell phenotype from exhausted to naive-like upon miR-155 KO, the expression of activation markers (e.g., *Tnfrsf9* and *Ifng*) was reduced in the KO mice, while memory/naive markers such as *Tcf7* and *Ccr7* were overexpressed (Fig. 3b). The KO TILs also scored lower in terms of the cycling signature (Fig. 3c). Altogether, these results are consistent with the critical role of miR-155 for T cell activation and differentiation. Unable to differentiate and acquire effector functions, miR-155 KO CD8+ TILs fail to control tumor growth[34]. In brief, by comparing the changing landscape and cell subtypes distribution of the KO

compared to the WT TILs, ProjecTILs provides a straightforward interpretation of the effect of this genetic alteration, all in the context of annotated, reference T cell subtypes.

While low-dimensional representations such as the UMAP are useful to summarize the most prominent transcriptional features that discriminate distinct T cell subtypes, they are often not sufficient to fully capture the heterogeneity of cell transcriptomes. With the goal to provide more resolution to the analysis of T cell states, and in particular to identify gene programs that are shared by multiple cell subtypes, we decomposed the reference atlas into 50 dimensions using Independent Component Analysis (ICA) (see "Methods"). Because ICA finds a representation of the data where the dimensions share minimal mutual information, it can be useful to separate different gene modules and transcriptional programs within complex gene expression datasets[35] and provides a complementary description to the UMAP representation—which in turn is built over a PCA reduction of the transcriptomic space. In order to interpret the biological relevance of the ICA components in the reference TIL atlas, we investigated their correlation with annotated molecular signatures from the mSigDB database[36] and observed several modules associated with key cellular pathways (Supplementary Fig. 4). For example, the component ICA 33 was driven by several genes associated with hypoxia (e.g., *Tpi1*, *Pkg1*, *Ldha*, *Slc2a1*); ICA 40 was rich in E2F targets (*Mcm2* to *Mcm7*, *Dut*, *Cdc6*), indicating a module of key regulators of cell cycle progression; ICA 26 contained multiple genes involved in cytotoxicity, such as *Prf1* (encoding perforin 1) and several granzymes (*Gzme*, *Gzmc*, *Gzmb*); and ICA 37 was strongly associated with response to interferons (e.g., *Ifit1*, *Ifit3*, *Rsad3*) (Supplementary Fig. 5). Critically, some ICA components appeared to affect mainly specific T cell subtypes or regions in the context of our reference map (e.g., ICA 43 and 45), while others captured features of multiple subtypes/regions (e.g., ICA 33 and 37, Supplementary Fig. 6).

Importantly, we can compare a projected query dataset with the reference map—or two query conditions between themselves—in their ICA representations, and identify components in which the query dataset deviates from the reference. ICA dimensions where the two sets differ significantly suggest gene modules that are up- or down-regulated in the query set, and may aid the biological interpretation of experimental observations. As an illustrative example of this approach, we re-analyzed the scRNA-seq from Wei et al.[37]. In this study, the authors show that ablation of *Zc3h12a* (which encodes Regnase-1) in CD8+ T cells improved the therapeutic efficacy of adoptively transferred, tumor-specific cells in mouse models of melanoma and leukemia. Projection of the Regnase-1-null CD8+ TILs and WT counterparts into our TIL reference atlas showed that cells from the two conditions occupied similar CD8+ regions of the map (Fig. 3d). However, Regnase-1-null CD8+ T cells displayed a 3-fold relative enrichment in Tpex and a 1.8-fold enrichment in Tex compared to WT T cells (Fig. 3e). Consistently, Regnase-1-null TILs showed an overall increased expression of *Tcf7*, *Slamf6*, and *Pdcd1* (Fig. 3f).

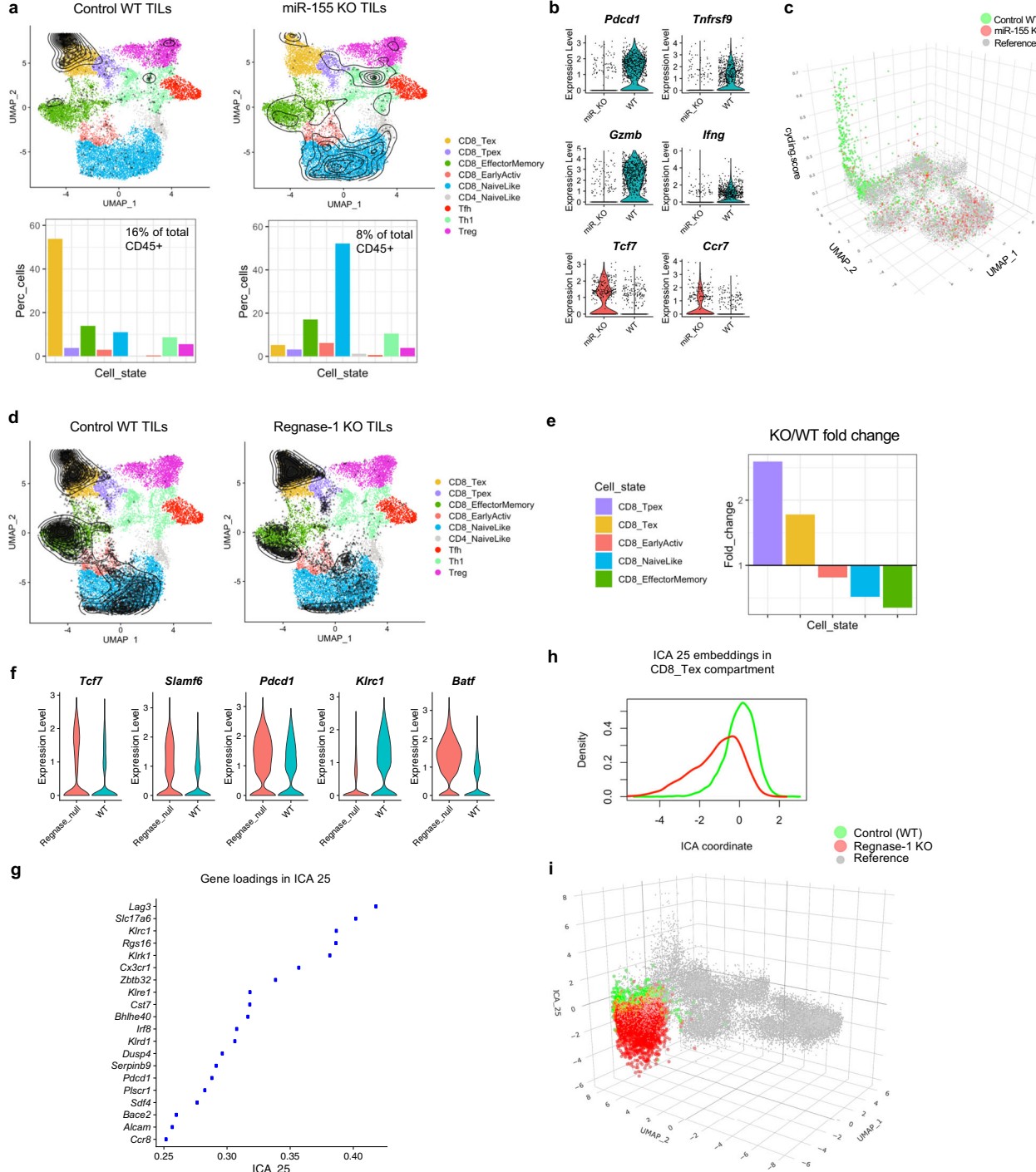

**Fig. 3 ProjecTILs reveals the effect of genetic perturbations on T cell transcriptomes and phenotypes. a–c** ProjecTILs analysis of the tumor CD45[+] scRNA-seq data by Ekiz et al.[34]: **a** WT and miR-155 KO TILs projected on the reference atlas (black points and density contours) and barplots depicting percentage of cells projected in each T cell subtype for the two conditions. T cells constituted 16% and 8% of the CD45[+] cells for the WT and miR-155 KO samples, respectively; (**b**) Violin plots showing expression of activation and cytotoxicity (*Pdcd1*, *Tnfsf9*/4-1BB, *Gzmb*, *Ifng*) and naive/memory (*Tcf7*, *Ccr7*) markers; (**c**) Cell cycling score represented on the *z* axis of the UMAP for the reference map of WT cells and miR-155 KO cells. **d–i** ProjecTILs analysis of the scRNA-seq data by Wei et al.[37]: **d** Single-cell projection on the reference TIL atlas (similar to A); (**e**) Fold-change in the Regnase-1 KO compared to WT for each TIL subtype containing at least 50 cells; (**f**) Global expression level for selected genes in the Regnase-1 KO versus the WT; (**g**) Top 20 driver genes in terms of gene loadings for the transcriptional program captured in ICA (Independent Component Analysis) component 25, the most discriminant dimension between the WT and the Regnase-1 KO; (**h**) Distribution of ICA 25 component for cells in the WT (green) versus Regnase-1 KO (red) samples, as can be also visualized (**i**) by plotting these values on the *z* axis of the UMAP plot.

Ablation of Regnase-1 favored acquisition of the Tpex state, explaining the increased T cell persistence reported by Wei et al. However, T cell-mediated tumor control additionally requires enhanced effector functions. To investigate whether Tex Regnase-1-null cells displayed altered gene programs compatible with an enhanced effector state, in addition to their increased relative abundance, we applied ProjecTILs discriminant ICA components analysis. Interestingly, ICA discriminant analysis of Tex cells between the two conditions revealed that the most significant deviation was in ICA 25 (KS test statistic = 0.612, $p = 0$), a component driven by the checkpoint molecules *Lag3* and *Klrc1* (Fig. 3g,h). Visualizing the ICA 25 coordinates of the Regnase-1-null cells on the *z* axis of the UMAP plot, we clearly observe that this component is highly down-regulated compared both to the WT data and the reference map (Fig. 3i). Accordingly, *Klrc1* expression was lower in Regnase-1-null cells (Fig. 3f). While analyzing the expression levels of key marker genes can suggest global patterns of alteration due to genetic perturbations (Fig. 3f), only by comparing gene programs in each cell subtype individually across conditions can one pinpoint the molecular programs affected by the perturbation while avoiding confounding effects due to global shifts in relative proportions of T cell subtypes. Importantly, a similar distribution of predicted cell subtypes and significant down-regulation of ICA 25 could be detected even in the absence of a control sample, by direct comparison of the Regnase-1-null cells to the reference map (Supplementary Fig. 7).

**Interpreting T cell states in the context of acute and chronic infections.** While ProjecTILs was originally conceived to study the diversity of TILs, it can be readily applied to any other biological context for which a reference atlas can be constructed. A prominent example is the lymphocytic choriomeningitis virus (LCMV) infection model, one of the best-studied models of acute and chronic viral infection. In order to construct a reference atlas for viral infection models, we collected scRNA-seq data of virus-specific CD8[+] T (P14) cells from three different studies[38–40], consisting of single-cell gene expression measurements at different time points for acute and chronic LCMV infection (Fig. 4a). Alignment by STACAS (Fig. 4b) was followed by unsupervised clustering (Fig. 4c, see "Methods"). By inspecting the gradients of gene expression across the UMAP representation of the atlas (Fig. 4d), as well as the average expression of a panel of marker genes in the different unsupervised clusters (Fig. 4e), we annotated seven functional clusters: effector early, effector intermediate, effector cycling, memory precursor, short-lived effector cells (SLEC), precursor exhausted (Tpex), and exhausted (Tex) (Fig. 4c). Cells from early acute infection (day 4.5) were mostly located in the early, intermediate and cycling effector areas, as well as in the memory precursor subtype; however, at a later time point (day 7.5) their distribution of cell subtypes shifted towards the SLEC subtype (Fig. 4f). Similarly, early chronic infection (day 4.5) was characterized by effector T cell types, nearly indistinguishable from the acute cells at the same time point; but as the infection progressed (days 7.5 and 30) their subtype distribution diverged from the acute infection towards Tpex and Tex subtypes (Fig. 4g) with a non-persistent wave of SLEC-like cells at an intermediate time point (day 7.5). An interactive interface to the viral CD8[+] T cell atlas is available at http://virustcellatlas.unil.ch.

With this virus-specific CD8[+] T cell reference atlas in hand, we proceeded to project new datasets to study the effect of genetic alterations on CD8[+] T cells during viral infection. The phosphatase PTPN2 has been proposed as an attractive immunotherapeutic target to enhance T cell cytotoxicity in chronic infection and cancer[39,41]. Automated ProjecTILs analysis

of co-transferred *Ptpn2*-KO and control P14 cells at day 30 after LCMV infection and CD4 depletion[39] revealed that the large majority of cells, both *Ptpn2*-KO and WT, were projected either in the Tpex or Tex clusters (Fig. 4h). Importantly, while the Tex compartment constituted only about 20% of the WT T cells, it amounted to over 50% of *Ptpn2*-KO cells (Fig. 4i). Moreover, analysis of the average expression of key marker genes confirmed a good agreement between Tpex expression profiles (for control, KO and reference subtype), as well as overexpression of *Pdcd1* and *Tox* in the *Ptpn2*-KO cells (Fig. 4j). Therefore, ProjecTILs automated analysis supported the original observations that in chronic infection, *Ptpn2* deletion promotes differentiation of Tpex into Tex, which translates into a higher number of effector cells, improving—at least transiently—viral and tumor control[39].

As a second example of projection, we applied ProjecTILs to study the effect of deleting the transcription factor *Tox* in virus-specific CD8[+] T cells during chronic viral infection. Projection of the data from Yao et al.[40] revealed that *Tox*-KO cells had a dramatic alteration in subtype composition compared to WT controls (Fig. 4k). In particular, it showed a large increase in the fraction of SLECs, at the expense of memory precursors and Tpex cells (Fig. 4l). Analysis of marker profiles confirmed that the *Tox*-KO cells classified as SLEC express high levels of *Klrg1* (Fig. 4m). These results are consistent with previous studies that demonstrated that TOX is required for the establishment of the exhaustion CD8[+] T cell program[22,40,42–44].

Finally, we projected P14 cells from CD4-depleted (or isotype control) chronically infected mice from Kanev et al.[45]. We observed that anti-CD4 antibody-treated mice had a dramatic shift in their T cell subtype composition from SLEC to Tpex compared to control mice, while the proportion of Tex remained similar (Supplementary Fig. 8a, b). This effect is in agreement with the conclusions of the original study, which found that in chronic infections, progenitor cells (Tpex) are unable to properly differentiate into effector cells in the absence of CD4[+] T cell help[45], and therefore "accumulate" in this state. We could also confirm that CD4 depletion had an effect on increasing *Pdcd1* expression, especially in Tex cells (Supplementary Fig. 8c). These data were generated using the SCRB-seq protocol[46], a type of sequencing that was not included in the reference LCMV atlas. Therefore, this example also highlights the robustness of ProjecTILs to accurately project, and enable the correct interpretation of, single-cell data across multiple sequencing platforms.

A crucial variable affecting the heterogeneity of T cells is their environment. Sandu et al.[47] investigated the diversity of CD8[+] T cells in chronic LCMV infection across six different tissues, and defined organ-specific transcriptomic profiles that could be divided into five main functional subtypes. Taking advantage of our reference CD8[+] T cell atlas for viral infection, we re-analyzed the data from Sandu et al. to investigate if tissue-specific transcriptomic alterations could be detected using our automated ProjecTILs pipeline (Fig. 5a). In general, the majority of virus-specific T cells were predicted to be terminally exhausted (Tex) as expected in this infection model, but different tissues were composed of variable fractions of other T cell subtypes (Fig. 5b). For instance, lung, blood, and spleen had the highest percentage of effector cells (SLEC), while lymph node and spleen had an exceeding percentage of Tpex cells compared to other tissues. While the original study defined fewer T cell states compared to our reference atlas for infection (Fig. 5c), the tissue-specific composition for the main T cell subtypes showed a remarkable correspondence between the ProjecTILs prediction and the original, unsupervised analysis (Fig. 5d). A unique advantage of projection into a stable reference atlas is that multiple samples (or multiple tissues, in this case) can be compared over the same

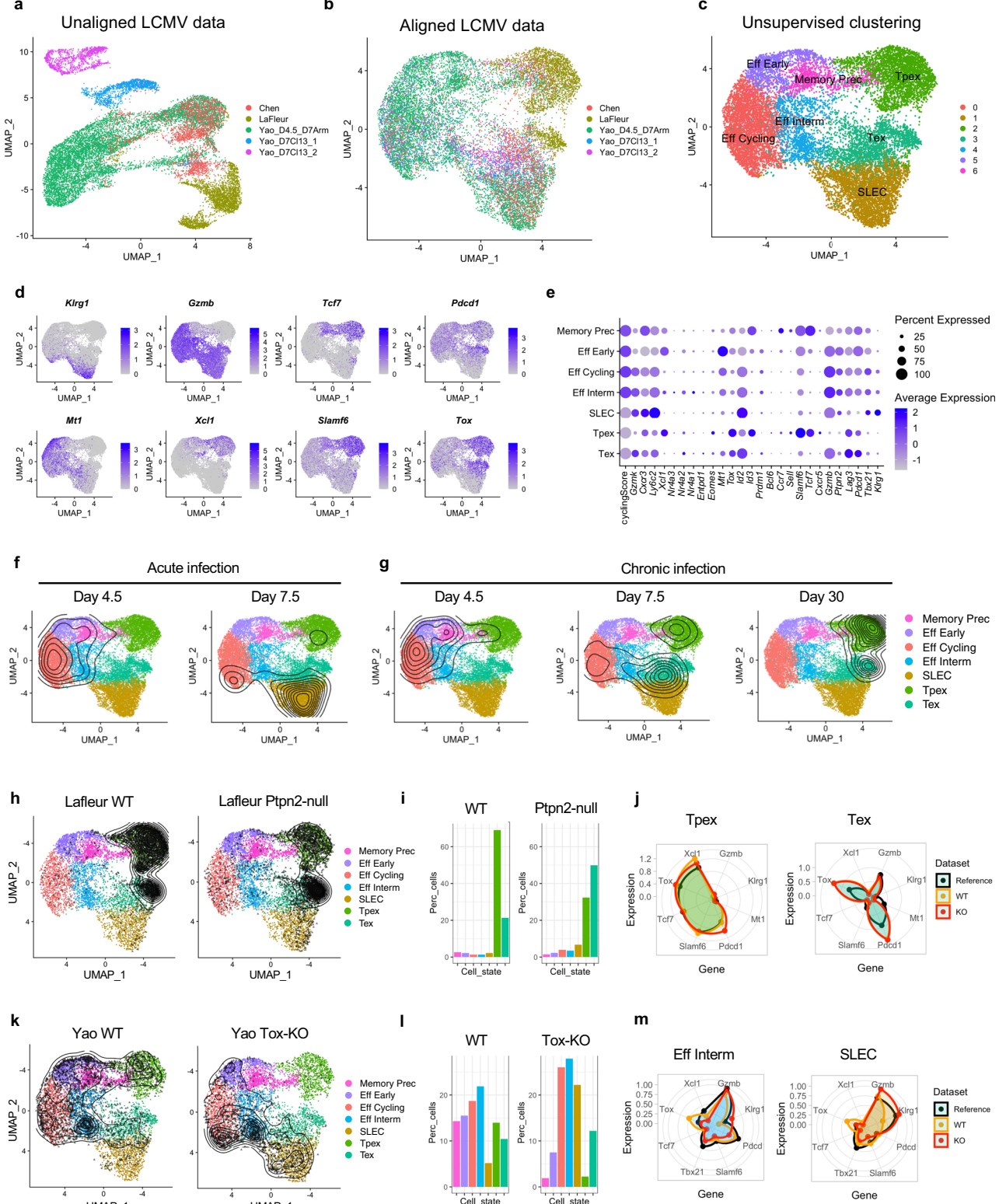

reference, and for specific T cell subtypes. Differential expression analysis between SLEC from blood and spleen showed that SLEC in spleen overexpress markers of activation such as *Nfkbia, Nr4a1,* and *Cd69* (Fig. 5e), indicating that these cells may have recently encountered antigen, unlike circulating cells. A similar observation can be made by comparing Tex cells from liver and spleen, but in this case also a significant overexpression of *Gzma* in liver is observed (Fig. 5f), as also noted in the original study.

Interestingly, two of the most discriminant ICA components between spleen and other tissues contain several genes involved in T cell activation and TCR signaling (Fig. 5g). This analysis demonstrates that, given an appropriate reference atlas, Projec-TILs can detect tissue-specific signals using a fully automated pipeline and default parameters, and obtain very similar results compared to a manually curated analysis performed by expert immunologists and bioinformaticians.

**Fig. 4 A reference atlas of virus-specific CD8+ T cells during acute and chronic infection. a** Unaligned datasets of lymphocytic choriomeningitis virus (LCMV)-specific CD8+ T (P14) cells during infection show pronounced batch effects, which (**b**) can be mitigated by STACAS alignment. **c** Unsupervised clusters were annotated to seven functional clusters by examining (**d**) the gradient of expression and (**e**) the average expression of marker genes by cluster, i.e., Memory Precursors; Early, Cycling, Intermediate, and short-lived (SLEC) effectors; Precursor Exhausted (Tpex) and Terminal exhausted (Tex) CD8+ T cells. **f** Density of cells across the map at two time points in acute infection and (**g**) at three different time points in chronic infection. **h–j** Analysis of *Ptpn2* KO versus control (WT) using the data by Lafleur et al.[39]: **h** ProjecTILs projection of WT and *Ptpn2* KO cells onto the infection reference map; (**i**) predicted percentage of cells for each T cell subtype; (**j**) normalized average expression for selected markers in the reference map, in WT and *Ptpn2* KO cells. **k–m** Analysis of *Tox* KO versus control (WT) using the data by Yao et al.[40]: **k** Projection in UMAP space by ProjecTILs for the WT and *Tox* KO samples; (**l**) predicted percentage of cells for each T cell type; (**m**) normalized average expression for selected markers in the reference map, in WT and *Tox* KO cells. Batches for integration (panels **a**, **b**): Chen: chronic infection day 8; LaFleur: chronic infection day 30; Yao_D4.5_D7Arm: acute and chronic infection day 4.5 + acute infection day 7; Yao_D7_Cl13_1 chronic infection day 7 sample 1; Yao_D7_Cl13_2 chronic infection day 7 sample 2.

**ProjecTILs reveals a strong conservation of TIL subtypes across species.** Mouse models are essential to gain mechanistic insights into tumor immune responses. Yet, precise definition of TIL subtype conservation between human and mouse has remained elusive. Here we asked whether the T cell states described in human tumors have a clear mapping to mouse TIL subtypes. Using orthologous genes between the two species, we applied ProjecTILs to analyze human TIL scRNA-seq data from 30 cancer patients from two cohorts (melanoma cohort of Li et al.[5] and basal cell carcinoma cohort of Yost et al.[48]) in the context of the reference murine TIL atlas (see "Methods"). Projected TILs from individual patients broadly distributed over the reference murine atlas (Fig. 6a, Supplementary Fig. 9).

ProjecTILs T cell subtype classification showed a good correspondence with the T cell annotations assigned by the authors of the original studies (Li et al.[5] and Yost et al.[48]). For example, human TILs originally defined as Treg, follicular-helper, naive or T-helper were largely projected to the corresponding subtypes on the murine reference atlas; the "CD8 effector" (Li et al.) and "CD8 cytotoxic" (Yost et al.) cells were projected to the reference effector memory (EM) subtype; the "CD8 exhausted" cells (Yost et al.) and "CD8 dysfunctional" cells (Li et al.), were mostly projected to the reference exhausted subtype (Fig. 6b, Supplementary Fig. 10). Surprisingly, a significant fraction of the cells annotated by the authors as "exhausted/dysfunctional" were projected on the EM state of the murine atlas. Further examination revealed that these cells displayed a clear effector-memory gene profile (i.e., high *GZMK, GZMA,* and *GZMB* expression) but lacked markers of exhaustion, such as *TOX, ENTPD1, HAVCR, PDCD1* (Fig. 6c), indicating that they were correctly projected to the EM reference subtype. SingleR[49] classification of exhausted/dysfunctional T cells of one cohort using expression profiles of the second cohort confirmed that a large fraction of these cells displayed a cytotoxic/effector-memory phenotype (Supplementary Fig. 11a, b). Similarly, naive cells from one cohort could not be unequivocally identified as such from the expression profiles of the other cohort (Supplementary Fig. 11c, d); a singleR model trained on human PBMC single-cell data[27] suggested that cells annotated as naive in both cohorts contained naive/central memory cells as well as CD4+ effector cells (Supplementary Fig. 11e, f), in line with the ProjecTILs classification.

As an alternative projection algorithm to interpret human TIL states, we applied Azimuth/Seurat 4[27], that by default utilizes a human PBMC reference atlas. We applied Azimuth to project T cell data from the Yost et al. cohort using the PBMC reference provided by the authors, as well as using our mouse TIL atlas. While broad T cell states (CD4+ vs. CD8+, Tregs) could be distinguished in the projections on the PBMC atlas, Azimuth could not discern between more specific subtypes. A considerable fraction of T cells were also assigned to the NK and MAIT areas of the reference map (Supplementary Fig. 12). When Azimuth was applied to the more specialized TIL reference atlas developed by us, projections appeared reasonable for certain cell types (Treg, Tfh, CD8+ effector), but naive-like cells were mostly misclassified, as well as exhausted T cells (Supplementary Fig. 13a, b). On a subset of cell subtypes that could be confidently mapped between studies, Azimuth resulted less accurate than ProjecTILs for the classification task (Supplementary Fig. 13c, d). These results highlight the importance of an accurate projection algorithm, but also of a robust atlas, specific for the problem at hand, rather than generalist, whole tissue atlases.

To further study the conservation of the reference TIL subtypes of our atlas, we analyzed scRNA-seq data from 132 tumor biopsies from 10 different studies, covering 7 different cancer types. After applying ProjecTILs projection and classification, we identified the differentially expressed genes of each TIL subtype across all datasets (see "Methods"), and used these to calculate average expression profiles for individual studies and TIL subtypes (Fig. 7, Supplementary Data 1). For example, *Gzma, Gzmk,* and *Ccl5* were significantly more expressed in CD8_EM cells from most cancer types, both murine and human; *Pdcd1, Havcr2,* and *Prf1,* among other genes, identified CD8+ exhausted cells; *Xcl1, Tnfsf4,* and *Tox,* among others, marked precursor exhausted CD8+ T cells; *Sell, Tcf7, Il7r,* and *Ccr7* were enriched in both CD4+ and CD8+ naive-like cells; *Foxp3, Il2ra, Ctla4* and several other genes identified Tregs; *Tox2* and *Tbc1d4* were differentially expressed in Tfh; and *Cd40lg, Anxa1* and *Rora* were enriched in T helper cells. Multiple other genes without previously documented associations with tumor-infiltrating T cells were also identified, revealing interesting targets for future validation (Supplementary Data 1). Overall, average expression profiles clustered preferentially by TIL reference subtype rather than by study, cancer type or species (Fig. 7). In particular, observing human and murine samples clustered together in each of the nine reference subtypes is statistically significant ($p < 3 \times 10^{-6}$), and points to a large conservation between human and mouse TIL states.

As a further example of projection to identify differences between human cohorts, we analyzed biopsies taken at baseline from the study by Sade-Feldman et al.[6], consisting of 19 melanoma patients that were classified as responders (R) and non-responders (NR) to checkpoint blockade. In agreement with the authors' observations, ProjecTILs revealed that *TCF7*-high TIL subtypes were enriched in responders vs. non-responders (Supplementary Fig. 14a). Intriguingly, these subtypes corresponded to naive-like CD8+ and CD4+ cells, and not to TILs displaying markers of tumor reactivity, such as PD-1[50]. A similar subtype bias was found between metastatic lymph nodes and tumors, irrespectively of the patients' responsiveness to immunotherapy (Supplementary Fig. 14b). This prompted us to analyze

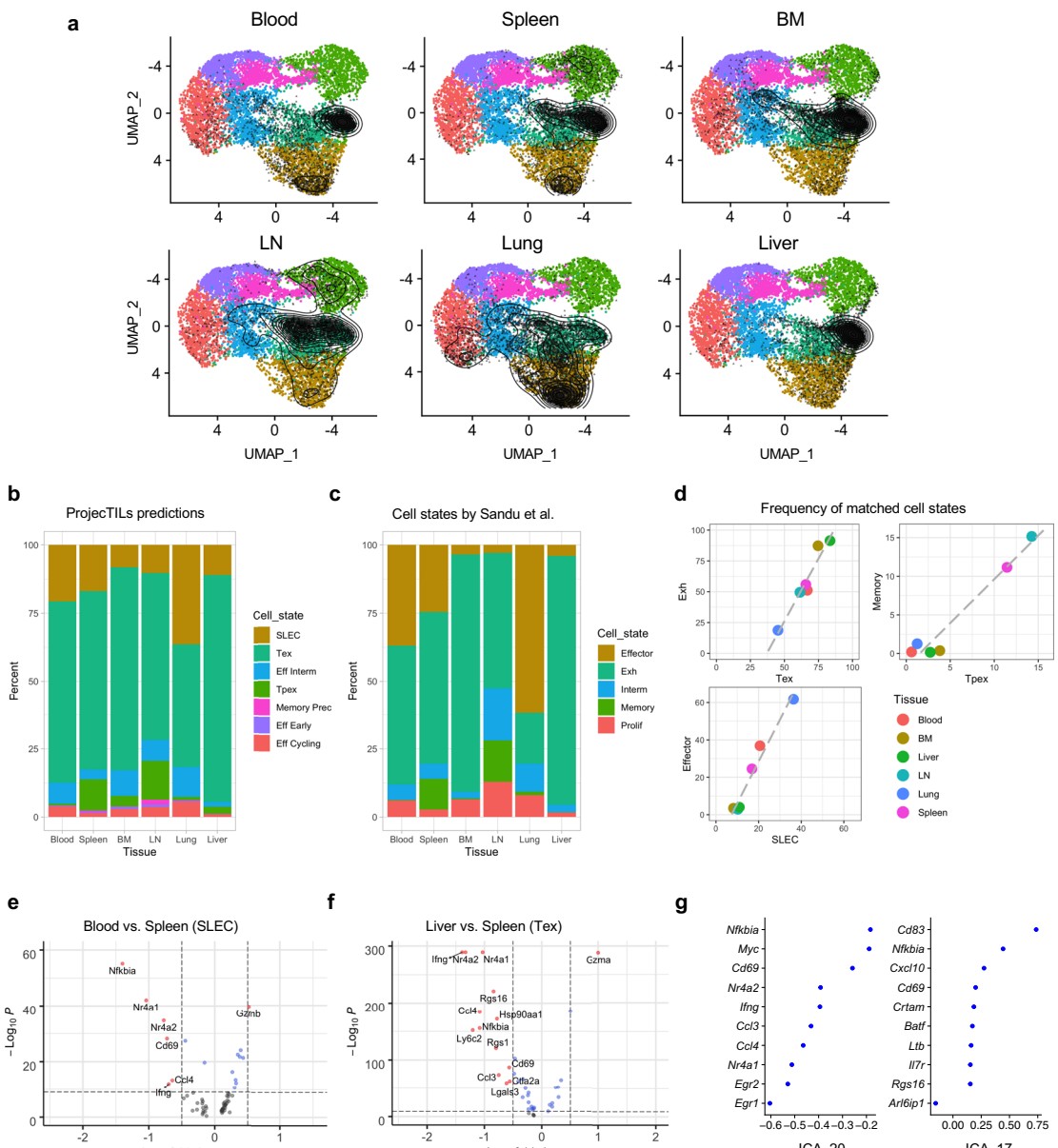

**Fig. 5 ProjecTILs resolves tissue-specific T cell heterogeneity in chronic viral infection. a** Projection of single-cell data onto the viral infection CD8+ T cell reference atlas for six different tissues, from the study by Sandu et al. **b** Distribution of ProjecTILs predicted T cells states for each tissue and (**c**) distribution of T cell states assigned in the original study by unsupervised analysis. **d** For three T cell states that could be confidently mapped between the original annotation and the ProjecTILs prediction (Exh ↔ Tex; Memory ↔ Tpex; Effector ↔ SLEC), the panels show the cell state percentage for each of the six tissues, according to the predicted (x axis) and original (y axis) cell annotation. **e** Volcano plot of differentially expressed genes between blood and spleen for cells predicted to be SLEC, and (**f**) between liver and spleen for cells assigned to the Tex state. **g** Two of the most discriminant ICA components (ICA 20 and ICA 17) between spleen and other tissues. *BM* bone marrow, *LN* lymph-node. Raw data for panel **c** courtesy of the authors of the original study.

in more depth the defining features of tumor-specific human TILs in terms of reference cell subtypes.

**Insights into the differentiation of human tumor-specific CD8+ T cells.** Some studies have suggested that only a fraction of tumor-resident CD8+ T cells are able to recognize tumor antigens[51,52] and identification of tumor-reactive T cells is far from trivial. Persistent antigenic stimulation of T cells in cancer and chronic infection induce a (TOX-driven) exhaustion program that sustains high expression of inhibitory receptors. Indeed, multiple surface markers associated with exhaustion have been

proposed as markers of tumor reactivity, including PD-1[50], TIM-3[53], and CD39[51].

To test the assumption that human Tex cells are tumor-specific, we first analyzed TIL subtype composition across tissues in the Li et al. melanoma cohort. As expected, the most abundant T cell subset in blood was the naive-like, followed by EM cells (Fig. 8a). Naive-like cells are likely to include both naive and central memory cells, which share very similar transcriptional profiles. Compared to blood, metastatic lymph node (mLN) biopsies were enriched in Tfh, Tregs, Tex, and Tpex, and tumor biopsies were strongly enriched in Tex and Tpex compared to mLN and blood (Fig. 8b). Moreover, we observed that the top

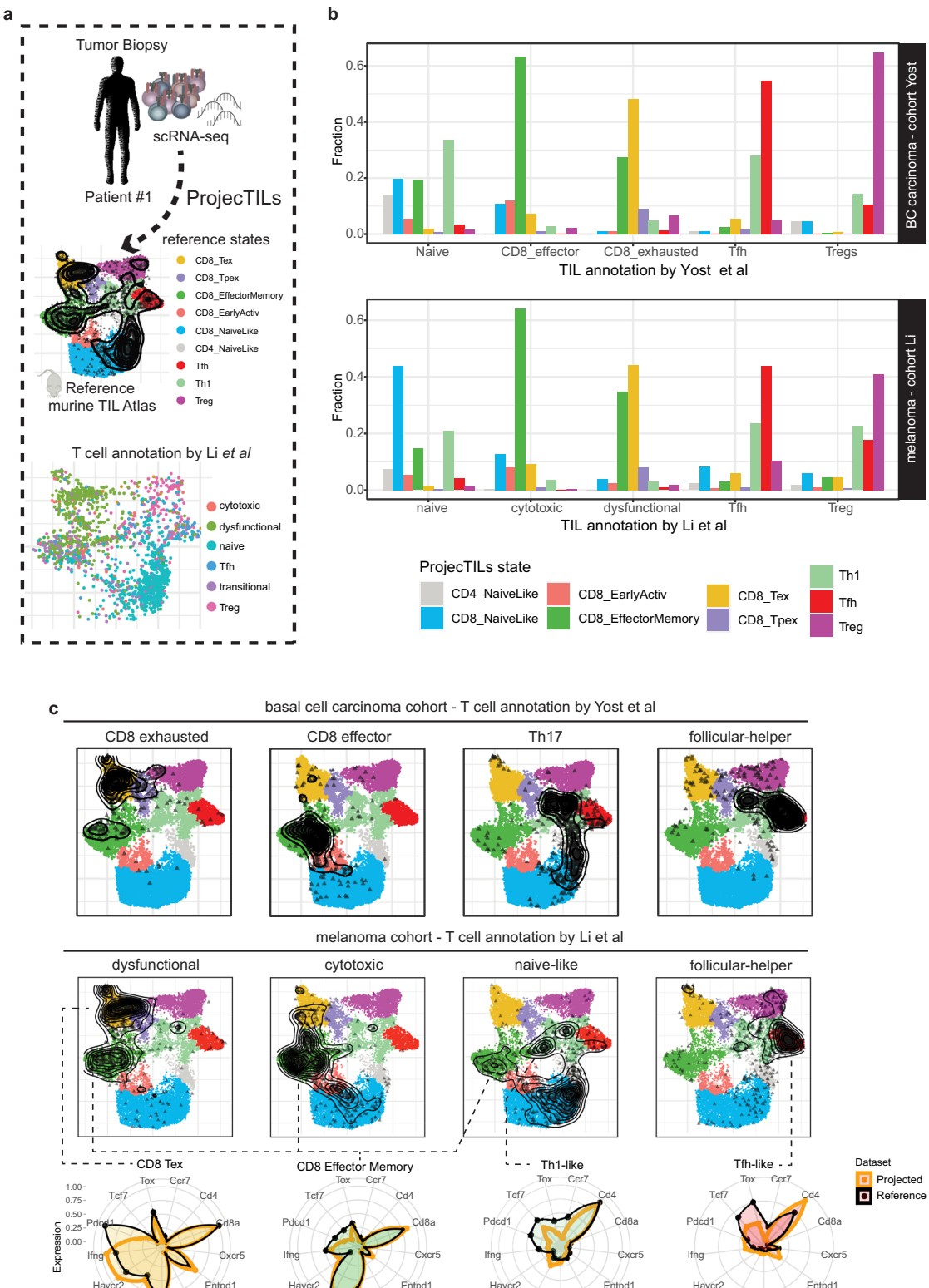

**Fig. 6 Accurate classification of human TIL states by projecting cancer patient transcriptomes on a reference mouse atlas. a** scRNA-seq data from patients' biopsies were analyzed using ProjecTILs in human-mouse orthology mode. Below, UMAP projection for TILs from one subject, colored by annotation according to Li et al. Projections for other subjects are available in Supplementary Fig. 9. **b** Fraction of cells classified in different subtypes by ProjecTILs compared to main original annotations by Yost et al. or Li et al. (complete annotation in Supplementary Fig. 10). **c** UMAP projections of cell subsets defined according to TIL state annotations by Yost et al. (e.g. exhausted, effector) or Li et al. (e.g. dysfunctional, cytotoxic). Radar plots display representative expression profiles of cells classified in the reference states for T cell marker genes. *BC carcinoma* basal cell carcinoma.

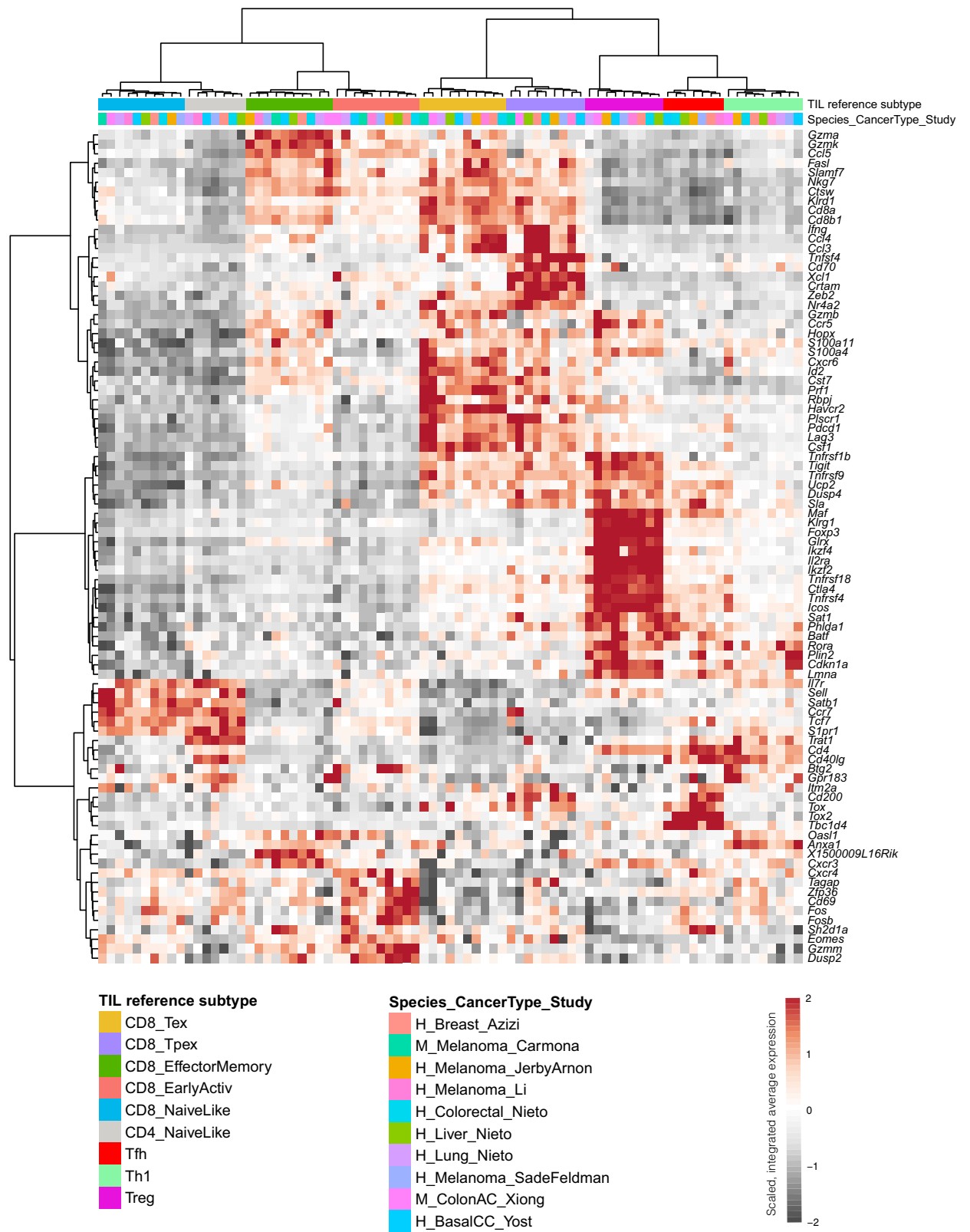

**Fig. 7 Conservation of T cell subtypes across studies, cancer types, and species.** Columns correspond to reference TIL subtypes for a given study, including all subtype-study combinations represented by at least 50 cells. Rows represent 88 marker genes, identified by concatenating all genes that were differentially expressed for at least one T cell state in at least four studies, limiting the number of genes per state to at most 25 genes. Values correspond to integrated average expression for the given gene and subtype-study combination, scaled and centered by row. Clustering by column shows that subtype-study expression profiles cluster preferentially by TIL subtype rather than by study, cohort or species (top colored bars). Species abbreviations: *H* human, *M* mouse.

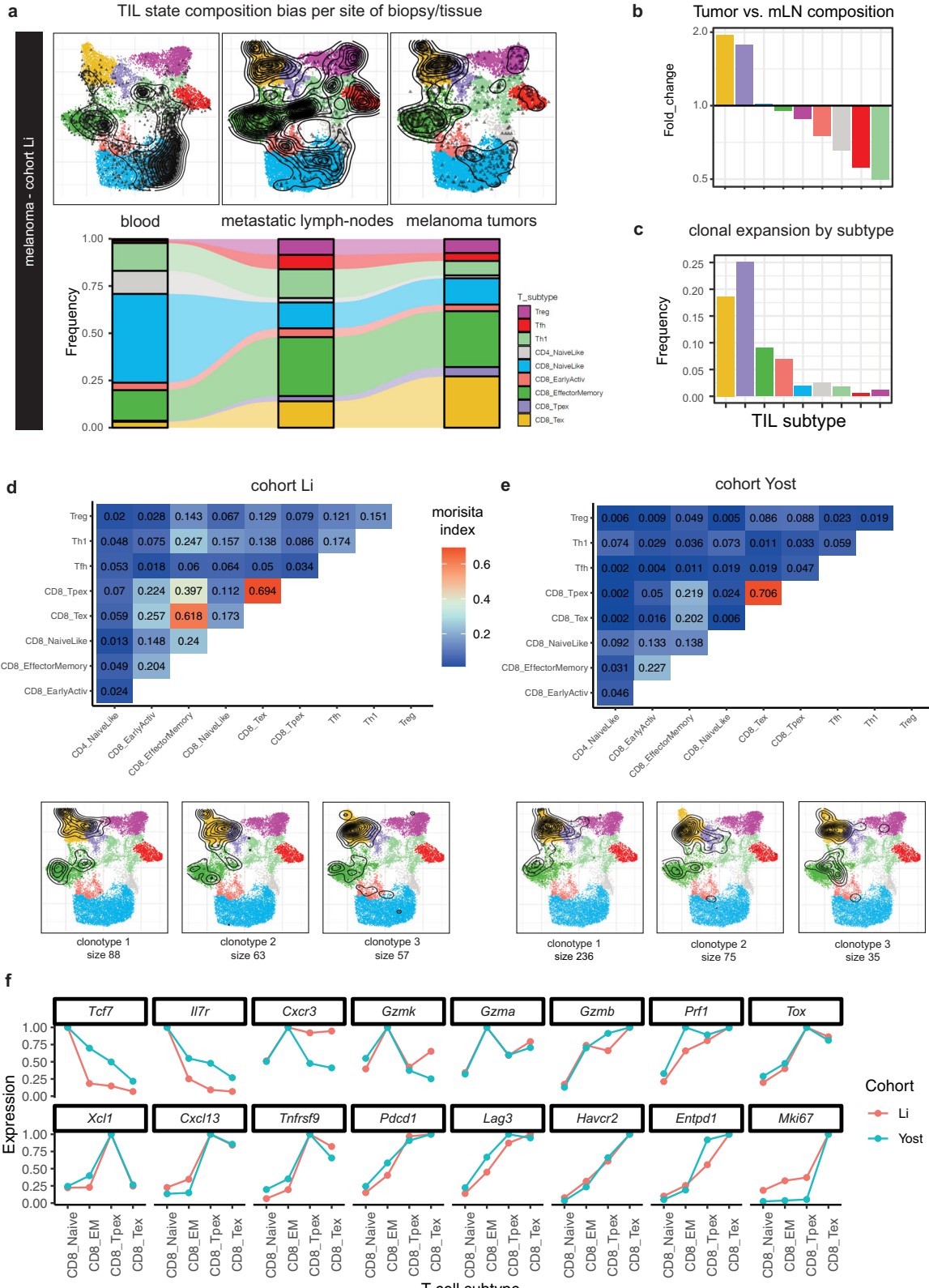

**Fig. 8 ProjecTILs analysis of human TIL states across tissues and their clonal relatedness. a** ProjecTILs projections and predicted subtype frequencies in biopsies from different tissues: blood, metastatic lymph nodes (mLN) and tumors (data from Li et al. cohort). **b** Subtype composition bias (fold change) in tumors vs mLN. **c** Frequency of cells from the top 10 expanded clonotypes over the total number of cells in each subtype. **d–e** The upper panels (heatmaps) display Morisita similarity indices measuring TCR repertoire overlap for each pair of TIL subtypes in the Li et al. (**d**) and Yost et al. (**e**) cohorts. Bottom panels: projection of TIL clones for the top three expanded clonotypes enriched in Tex or Tpex subtypes in each patient cohort. **f** Average normalized gene expression of human T cells projected in the CD8+ NaiveLike, CD8+ Effector Memory (CD8_EM), CD8+ Tpex and CD8+ Tex subtypes for a panel of key marker genes.

expanded clonotypes were mostly occupying the Tex and Tpex subtypes (Fig. 8c). The positive correlation of Tex and Tpex subtype frequency with tumor burden, as well as the enrichment of clonally expanded T cells in these compartments, is consistent with the notion that, in cancer patients, tumor-specific TILs are mostly found in (*PDCD1*-high *TOX*-high) exhausted states, similarly to mouse[28,31].

We next exploited TCR clonal linkage to evaluate the presence of tumor-specific TILs that do not correspond to Tex and Tpex subtypes. First, we calculated TCR clonal repertoire overlap between TIL subtypes. We verified a strong overlap between Tex and Tpex repertoires in both cohorts, as measured by Morisita similarity index (Fig. 8d,e, top panels), consistent with mice studies showing that Tpex cells give rise to Tex[21,28,31,54]. Interestingly, we also found a similarly strong clonal relatedness between Tex/Tpex and EM subtypes in the Li cohort as well as, to a lesser extent, in the Yost cohort (Fig. 8d,e). This suggested that a fraction of EM TILs were also tumor specific. Next, we selected all clonotypes that were enriched (at least 50% of the clones) in Tex or Tpex—i.e., tumor-specific clonotypes. Projection of these tumor-specific clonotypes confirmed that they spanned the three Tex, Tpex, and EM subtypes in the two cohorts (Fig. 8d–e, bottom panels).

Gene expression profiles for human T cells projected onto the reference murine atlas confirmed that most key T cell markers were consistent with their ProjecTILs subtype assignment (Fig. 8f). *TOX* and *TNFRSF9* (4-1BB) expression values were higher in Tpex and Tex compared to EM, indicating higher exhaustion and activation levels in Tpex and Tex. Consistently, *PDCD1*, *LAG3*, *HAVCR2* (TIM-3), and *ENTPD1* (CD39) were also higher in Tex compared to EM, and lower in EM compared to the naive-like state. In contrast, *CXCR3*, *GZMK*, and *GZMA* expression was highest in EM. Compared to Tex, tumor-specific Tpex cells expressed higher levels of *TCF7* and *IL7R*, and lower levels of cytotoxicity molecules including *GZMA, GZMB*, and *PRF1*. Notably, expression of the type 1 classical dendritic cells (cDC1) chemoattractant XCL1 was specific to the Tpex subtype, consistent with Tpex gene profiling in mice[21,28,31] and their co-localization with professional antigen-presenting cells niches in murine and human tumors[31,55]. Finally, Tpex had lower expression of cell cycling genes such as *MKI67* compared to Tex, consistent with their higher quiescence.

Altogether, these observations demonstrate that, in different human cancer types, tumor-specific CD8+ TILs co-exist in three distinct subtypes: a cytotoxic *TOX*-high exhausted subtype; its *TOX*-high *TCF7*-high exhausted precursor, quiescent and characterized by lower cytotoxicity; and a precursor subtype that does not display the hallmarks of tumor-specific TILs but resembles blood-circulating effector memory T cells. These results are compatible with a model in which *CXCR3*-high blood-circulating EM cells are recruited in the tumor, irrespectively of their antigen specificity. Then, rare tumor-specific *TOX*-low EM TILs driven by persistent antigenic stimulation differentiate into *TOX*-high *XCL1*-high quiescent (Tpex) cells which, following interaction with XCR1+ APCs, give rise to highly proliferative terminally exhausted/dysfunctional (Tex) CD8+ TILs that engage in tumor cell killing (Fig. 9). Alternatively, TCR clonal linkage is compatible with a model in which some EM TILs might directly differentiate into Tex cells, without transitioning through the Tpex subtype. Importantly, our results demonstrate that these subtypes are conserved across cohorts, cancer types, and species.

## Discussion
We share the goal of many others to be able to "read" immunological states in health and disease by single-cell transcriptomics

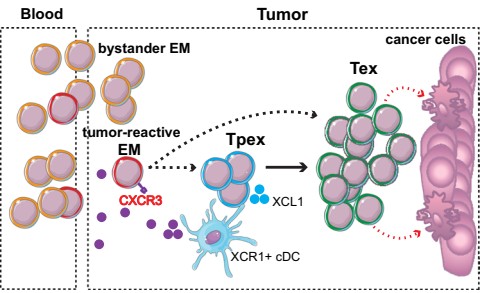

**Fig. 9 A model of intratumoral CD8+ T cell differentiation supported by meta-analysis of human scRNA-seq data using ProjecTILs.** Blood-circulating CXCR3-high EM cells are recruited to the tumor; these include tumor-specific EM cells as well as bystander TILs. Persistent antigen stimulation drives differentiation of tumor-specific EM TILs into XCL1-high Tpex cells which, following interaction with XCR1+ APCs, give rise to highly proliferative Tex CD8+ TILs with capacity to kill cancer cells. An alternative differentiation path from EM directly to Tex is also plausible. EM: CD8+ effector memory/CD8_EM (*TOX*-low *GZMK*-high *CXCR3*-high). Tpex: CD8+ precursor-exhausted (*TOX*-high *TCF7*-high *GZMB*-low). Tex: CD8+ exhausted/dysfunctional (*TOX*-high, *TCF7*-low, *GZMB*-high).

and identify therapeutic opportunities. Definition of robust, biologically relevant cellular states by scRNA-seq analysis is typically an iterative, time-consuming process that requires advanced bioinformatics and biological domain expertise. Even after successful analysis, cell clusters are not directly comparable between studies, preventing us from learning general biological rules across cohorts, conditions, and models. Construction of single-cell transcriptomic atlases is a very effective approach to condense the diversity of molecular profiles within a cell type, a tissue, or an entire organism. This is particularly powerful to characterize the heterogeneity of cell populations, to elucidate mechanisms and trajectories of differentiation as well as to aid the design of therapies targeted to specific cell types. Reference atlases can also serve as a reliable, stable baseline for the interpretation of new experiments, against which to evaluate the effect of cellular perturbations, such as changes in the balance between pre-existing subtypes or the identification of novel states in response to immunotherapies. In this work we described a computational method to interpret immunological states by projecting scRNA-seq data onto a reference T cell atlas, allowing the analysis of new data in the context of a stable, curated collection of T cell subtypes. Compared to other available methods, ProjecTILs has the advantage that its reference atlas remains unaltered upon projection of new datasets. Therefore, this approach allows mapping into the same reference space T cell states that were defined across different studies, cohorts and cancer types, and provides a framework for large scale meta-analyses to identify cell states associated with prognosis and responsiveness to immunotherapy.

While it is useful to summarize cell heterogeneity as discrete subtypes for conceptualization and the design of experimental validations, it is also apparent that cells exist in a continuum of states, which would be best described as probability distributions, or regions in a multi-dimensional transcriptional space. A key advantage of embedding new data into a reference atlas of cell states is that the query cells can be interpreted in a continuous space of transcriptional states, allowing the visualization of dynamic changes between experimental conditions. While ProjecTILs can be thought of as a classifier into pre-annotated discrete reference states, it also operates in a continuous space and can, therefore, capture intermediate and transient cellular states. For instance, we observed a pattern of expression of the chemokine receptor *Cx3cr1* that straddles the Tpex and Tex subtypes in the TIL atlas (Fig. 1f), as well as a gradient of expression for this

molecule within the Tex compartment in infection. Indeed, multiple studies have identified *Cx3cr1* as a marker for a transitory, intermediate state between the Tpex and Tex populations[56–58].

Beyond the interpretation of new data in terms of known, annotated cell subtypes, we have shown that ProjecTILs can aid the discovery of novel states that deviate from the reference, e.g., as a consequence of a genetic alteration or tissue adaptation. A case in point, analysis of the Regnase-1 KO data of Wei and colleagues not only explained the increased T cell persistence and tumor control in terms of changes in T cell subtype frequencies (Fig. 3e) but also revealed the previously unreported down-regulation of a novel inhibitory gene program in exhausted CD8+ TILs. This program (ICA 25) was driven by *Klrc1* (coding NKG2A) and *Lag3* (Fig. 3g,h), and its expression was uncoupled from that of other inhibitory receptors such as PD-1, TIM-3, or CTLA4, suggesting a potential benefit of targeting these two programs simultaneously. Indeed, dual blockade of PD-1 and LAG-3 results in robust and synergistic reinvigoration of Tex cells in cancer[59] and chronic infection[60], while NKG2A blockade has been shown to potentiate CD8+ T cell immunity induced by cancer vaccines[61].

In most experiments aimed at studying the effect of a perturbation, it is imperative to design a control group, as a baseline to compare the effect of the perturbation of interest. In single-cell experiments, when studying the effect of a genetic perturbation or of a treatment, the control group usually consists of the same cell population as the perturbation group but under basal conditions. It is conceivable, as reference atlases become increasingly complete, that the control group is already satisfactorily included in the reference atlas—the new condition could then be directly evaluated against the atlas, bypassing the need for deeply re-sampling the transcriptional space of basal conditions, as we have illustrated with the analysis of Regnase-1 KO data in absence of a control sample (Supplementary Fig. 7).

Compared to murine models, the analysis of human TIL states is complicated by the large genetic and environmental variability between patients, as well as the large variability between biopsies due to tissue-specific effects and tumor heterogeneity. Tumor scRNA-seq studies typically describe T cell heterogeneity in terms of several clusters, which are then manually annotated in "states" or "subtypes" defined by the authors and that tend to suffer from batch effects between samples. As a result, systematic comparison of T cells states across studies, cohorts and cancer types becomes extremely difficult. In this work, by meta-analysis of 132 cancer patient biopsies, we have shown that ProjecTILs can accurately project human T cell transcriptomes onto a reference mouse atlas, and that human TIL heterogeneity can be largely explained in terms of robust T cell subtypes. Such level of conservation between human and mouse TIL states is encouraging for translational research in cancer immunotherapy. It also provides the foundations to identify and characterize human-specific T cell heterogeneity.

Preventing or reverting exhaustion/dysfunction of tumor-specific CD8+ T cells is currently one of the major goals in cancer immunotherapies. While there is evidence suggesting that pre-exhausted/dysfunctional tumor-specific CD8+ T cells are present in human tumors, a robust definition of such TIL states and the differentiation process by which they acquire exhaustion features has remained elusive. Our meta-analysis of scRNA-seq and TCR-seq data from two cohorts of melanoma and basal cell carcinoma patients with ProjecTILs, revealed that (i) the majority of human TILs do not display features of exhaustion or tumor-reactivity, and are clonally disconnected from the exhausted TILs, suggesting that most of them are not tumor specific; and that (ii) tumor-specific exhausted/dysfunctional CD8+ TILs can co-exist with two rare, quiescent precursor subtypes: an exhausted TOX+

PD1+ TIM3- XCL1+ (Tpex) state; and a pre-exhausted/dysfunctional (EM) state with low expression of *TOX* and inhibitory receptors, and high expression of *CXCR3* and *GZMK*, that resemble blood circulating EM cells. The proposed CD8+ TIL differentiation model based on these observations has important implications for the design of therapies aimed at preventing T cell exhaustion, and for the identification of tumor-specific T cells with high stemness for their use in adoptive cell therapies[62].

We have described the construction of reference single-cell atlases for murine T cells in pan-cancer and infection models that are strongly supported by literature. While we observed that the main, known T cell subtypes can be accurately recapitulated in these reference maps, they do not yet encompass the full diversity of transcriptional states that can be acquired by T cells, especially for CD4+ TILs (which were under-represented compared to CD8+ among available data) and for γδ T cells (which were not represented at all). Only very recently, with the popularization of single-cell technologies, it has become possible to generate data of sufficient depth and quality to construct such high-resolution reference maps. We are therefore just beginning to appreciate the full potential of combining information from multiple studies and perform meta-analyses across models, tissues, and cancer types. Considering the pace at which new single-cell data are generated, we anticipate that reference maps will quickly grow in size and completeness, increasingly covering the space of possible transcriptional states that can be assumed by individual cells. We expect that the accuracy of projection of new data into such exhaustive reference atlases will also improve as a consequence.

While we have shown that ortholog mapping offers a viable solution to interpret human T cell responses in the context of a robust mouse atlas, we envision that, with rapidly growing data, it will soon become feasible to construct high-quality reference human T cell atlases able to capture human-specific diversity. In this respect, mouse atlases could serve as scaffolds to build their human counterparts. Finally, projecting whole tissue data into a collection of high-resolution cell type atlases, covering multiple immune cell compartments, would enable interpreting immune responses at a systems level, by the study of correlation, and putative interaction and cross-talk, between not only cell types, but between cells in very specific differentiation states.

We have implemented ProjecTILs as an R package (https://github.com/carmonalab/ProjecTILs) and we provide a Docker image ready to use. Because ProjecTILs is integrated with Seurat, it can be easily combined with other tools for up- and down-stream analyses. We believe our approach will have a great impact in revealing the mechanisms of action of experimental immunotherapies and to guide novel therapeutic interventions in cancer and beyond.

## Methods

**Mice**. Eight- to 10-week-old female C57Bl/6 mice were obtained from The Charles River Laboratories and housed at Genentech in standard rodent micro-isolator cages to be acclimated to study conditions for at least 3 days before tumor cell implantation. Mice were housed in individually ventilated cages within animal rooms maintained on a 14:10-h, light:dark cycle. Animal rooms were temperature and humidity-controlled, between 68–79 °F and 30–70% respectively, with 10–15 room air exchanges per hour.

All animal studies were reviewed and approved by Genentech's Institutional Animal Care and Use Committee. Mice were maintained under specific pathogen free conditions under the guidelines of US National Institute of health. Genentech is an AAALAC-accredited facility and all animal activities in the research studies were conducted under protocols approved by the Genentech's Institutional Care and Use Committee (IACUC). Mice whose tumors exceeded acceptable size limits (2000 mm3) or became ulcerated were euthanized and removed from the study.

**Single-cell RNA-seq of tumor-draining lymph node T cells**. Tumor draining lymph nodes from mice with established MC38 tumors (~190 mm³) were excised and single-cell suspensions were stained for CD45 (Biolegend, clone 30-F11,

dilution 1:100), TCRb (Biolegend, clone H57-597, 1:100), CD44 (eBioscience, clones IM7, 1:200), CD62L (eBioscience clone Mel14, 1:80) and LIVE/DEAD (Life Technologies, Fixable Dead Cell Stain) and sorted into CD45$^+$TCRb$^+$ T cells, gating out the antigen-inexperienced CD62L$^+$CD44$^-$ population. Sorted cells were then loaded onto a 10x Chromium Chip A using reagents from the Chromium Single-Cell 5′ Library and Gel Bead Kit (10x Genomics) according to the manufacturer's protocol. Amplified cDNA was used for both 5′ RNA-seq library generation and TCR V(D)J targeted enrichment using the Chromium Single-Cell V(D) J Enrichment Kit for Mouse T Cells (10x Genomics). 5′ RNA-seq and TCR V(D)J libraries were prepared following the manufacturer's user guide (10x Genomics). The final libraries were profiled using the Bioanalyzer High Sensitivity DNA Kit (Agilent Technologies) and quantified using the Kapa Library Quantification Kit (Kapa Biosystems). Each single-cell RNA-seq library was sequenced in one lane of HiSeq4000 (Illumina) to obtain a minimum of 20,000 paired-end reads (26 × 98 bp) per cell. Single-cell TCR V (D)J libraries were multiplexed and sequenced in one lane of HiSeq2500 (Illumina) to obtain minimum of 5000 paired-end reads (150 × 150 bp) per cell. The sequencing specifications for both single-cell RNA-seq and TCR V(D)J libraries were according to the manufacturer's specification (10x Genomics).

Single-cell RNA-seq data for each replicate were processed using cellranger count [CellRanger 2.2.0 (10x Genomics)] using a custom reference package based on mouse reference genome GRCm38 and GENCODE[63] gene models. Individual count tables were merged using CellRanger *aggr* to reduce batch effects.

**Cell lines**. The murine colon adenocarcinoma MC38 cell line was obtained from a former Genentech colleague, Rink Offringa, in 2008. Cells were cultured in RPMI-1640 medium plus 2 mmol/L l-glutamine with 10% fetal bovine serum (HyClone). Cells in log-phase growth were centrifuged, washed once with Hank's balanced salt solution (HBSS), counted, and resuspended in 50% HBSS and 50% Matrigel (BD Biosciences) at $1 \times 10^6$ cells/mL for injection into mice.

**Batch-effect correction and construction of reference atlases**. Prior to dataset integration, single-cell data from individual studies were filtered using TILPRED-1.0 (https://github.com/carmonalab/TILPRED), which removes cells not enriched in T cell markers (e.g., *Cd2, Cd3d, Cd3e, Cd3g, Cd4, Cd8a, Cd8b1*) and cells enriched in non-T cell genes (e.g., *Spi1, Fcer1g, Csf1r, Cd19*). Dataset integration was performed using STACAS[14] (https://github.com/carmonalab/STACAS), a batch-correction algorithm based on Seurat[12]. For the TIL reference map, we specified 600 variable genes per dataset, excluding cell cycling genes, mitochondrial, ribosomal, and non-coding genes, as well as genes expressed in <0.1% or >90% of the cells of a given dataset. For integration, a total of 800 variable genes were derived as the intersection of the 600 variable genes of individual datasets, prioritizing genes found in multiple datasets and, in case of draws, those derived from the largest datasets. We calculated pairwise dataset anchors using STACAS with default parameters, and filtered anchors using an anchor score threshold of 0.8. Integration was performed using the IntegrateData function in Seurat, providing the anchor set identified by STACAS, and a custom integration tree to initiate alignment from the largest and most heterogeneous datasets. Similarly, to construct the LCMV reference map, we split the datasets into five batches that displayed strong technical differences, and applied STACAS to mitigate their confounding effects. We computed 800 variable genes per batch, excluding cell cycling genes, ribosomal and mitochondrial genes, and computed pairwise anchors using 200 integration genes, and otherwise default STACAS parameters. Anchors were filtered at the default threshold 0.8 percentile, and integration was performed with the IntegrateData Seurat function with the guide tree suggested by STACAS.

Both for the TIL and LCMV atlases, we performed unsupervised clustering of the integrated cell embeddings using the Shared Nearest Neighbor (SNN) clustering method[64] implemented in Seurat with parameters {resolution = 0.6, reduction = "umap", k.param = 20} for the TIL atlas and {resolution = 0.4, reduction = "pca", k.param = 20} for the LCMV atlas. We then manually annotated individual clusters (merging clusters when necessary) based on several criteria: (i) average expression of key marker genes in individual clusters; (ii) gradients of gene expression over the UMAP representation of the reference map; (iii) gene-set enrichment analysis to identify over- and under- expressed genes per cluster using MAST[65]. In order to have access to predictive methods for UMAP, we recomputed PCA and UMAP embeddings independently of Seurat using respectively the *prcomp* function from basic R package "stats", and the "umap" R package (https://github.com/tkonopka/umap).

**The ProjecTILs pipeline**. The essential input to the ProjecTILs pipeline is an expression matrix, where genes are rows and cells are columns. If raw counts (e.g., UMI counts) are provided, each entry $x$ in the matrix will be normalized using the formula: $\log (1 + 10{,}000\, x / S)$, where $S$ is the sum of all counts for that cell, and log is the natural logarithm. To ensure that only T cells are included in the query dataset, by default TILPRED-1.0 is applied to predict the composition of the query, and all cells annotated as "Non-T cells" or "unknown" are removed from the query. This filter can be optionally disabled by the user. Then, a reference atlas of annotated cells states (by default the TIL atlas) is loaded into memory, together with its cell embeddings in gene, PCA and UMAP spaces, and all associated

metadata. In order to bring the query data in the same representation spaces as the reference map, batch-effect correction is applied to the normalized cell-gene counts of the query set using the anchor-finding and integration algorithms implemented in STACAS and Seurat, where the genes for integration consist of the intersection of the variable genes of the reference map and all genes from the query. After batch-effect correction, the PCA rotation matrix pre-calculated on the reference atlas (i.e., the coefficients allowing the transformation from reference gene space into PCA space) is applied to the normalized, batch-corrected query matrix. In the same way, the predict function of the "umap" package allows transforming PCA embeddings into UMAP coordinates. By this means, the query data can be embedded into the original, unaltered coordinate spaces of the reference atlas, enabling joint visualization as well as classification of the query cells into T cell subtypes.

ProjecTILs is implemented as a modular R package, with several functions that aid interpretation and analysis. The make.projection function is the core utility that implements the projection algorithm described above. It can be run in "direct" mode, in which case the PCA and UMAP rotations are directly applied without batch-effect correction. This may be useful for very small datasets, where alignment and integration algorithms will not be applicable. To project human data onto a murine reference atlas, the user must set the flag "human.ortho = TRUE", which automatically converts human genes to their mouse orthologs before projection. Plot.projection allows visualizing the query dataset as density level curves superimposed on the reference atlas. The cellstate.predict function implements a nearest-neighbor classifier, which predicts the state of each query cell by a majority vote of its annotated nearest neighbors (either in PCA or UMAP space) in the reference map. Find.discriminant.genes performs differential expression analysis for specific cell states/subtypes between two paired conditions, or alternatively between one condition and the reference map. Find.discriminant.dimensions analyses PCA and ICA embeddings (described below) to identify dimensions where the query deviates significantly from the reference map. Several additional functions allow visualizing multiple aspects of the reference and projected dataset and aid the biological interpretation of the results. The code and description of the package, together with tutorials and applications to analyze public datasets can be found at: https://github.com/carmonalab/ProjecTILs.

**ICA and discriminant dimensions**. Independent component analysis (ICA) is a computational technique aimed at deconvoluting a multivariate signal (such as simultaneous expression of many genes) into additive, independent sources (in this case different genetic programs). Because ProjecTILs relies on PCA for dimensionality reduction and projection, we reasoned that ICA components could provide a complementary and non-redundant decomposition of transcriptomics signals compared to PCA. We applied the fastICA implementation[66] to calculate 50 independent components on the integrated expression matrix of the TIL reference atlas. To suggest a biological interpretation of the ICA components, we downloaded hallmark gene sets (H) and canonical pathway gene sets (CP) from the Molecular Signatures Database[36] (mSigDB), as well as selected immunological signatures from previous studies. We scored each ICA against these signatures by summing the ICA gene loadings for all genes in a given signature, and then taking the absolute value of this score. We retained the top-three scoring signatures for each ICA, and clustered ICA components based on the union of all retained signatures (see Supplementary Fig. 4).

After projection, query datasets are also subject to transformation in ICA space through the ICA rotation matrix. The find.discriminant.dimensions function implemented in ProjecTILs evaluates the distribution of the cells in the query for each ICA dimension, and compares it to the distribution of cells in the reference (or to a control query dataset, if provided) in the same ICA dimension. For each ICA dimension, a statistical test can then be applied to confirm or reject the null hypothesis that, in this dimension, the cells in reference and query are drawn from the same distribution. ProjecTILs implements a Kolmogorov-Smirnov (KS) test or a $t$ test to the null hypothesis, multiplying $p$ values by the number of tests (i.e., 50) to correct for multiple testing (i.e., Bonferroni correction). ICA dimensions where the query deviates significantly from the reference (or the control, if provided) are ranked by their test statistic to identify the top discriminant dimensions. ICA embeddings can be visualized as an additional dimension on the $z$ axis of the reference UMAP space with the function plot.discriminant.3d.

**Cross-validated projection benchmark**. To estimate the accuracy of the projection algorithm, we devised a cross-validation experiment where we removed part of the data from the reference before projecting the removed data back into the map. Each of the seven datasets included in the reference TIL map, except one, was composed of at least two samples; we constructed a cross-validation experiment by removing, at each cross-validation step, half of the samples of a given dataset, and then projected these samples into a reduced version of the map that does not contain the data points from these samples. The dataset by Singer et al.[67] consists of a single sample, and therefore we removed all of its cells before projecting it back into its reduced map. After systematically projecting all cells in cross-validation, we compared their projected coordinates (either in UMAP or PCA space) with their original coordinates in the reference map (Supplementary Fig. 2). By the same token, we evaluated the performance of ProjecTILs (and two other methods, with parameters detailed in the next section) by comparing the composition of any given

dataset in terms of predicted cell states compared to the annotated cell states in the reference map (Supplementary Fig. 3). We found that using ProjecTILs with batch-correction, 90.0% of the projected cells are found within a radius of one unit (their neighborhood) in UMAP space (Supplementary Fig. 2) from their original coordinate in the reference map, and 93.4% within a radius of five units in PCA space. In terms of cell state classification, 91.6% of the projected cells were correctly assigned to their cell subtype (Supplementary Fig. 3a).

**Projection with Azimuth and scMap.** By default, Azimuth runs on a reference atlas of human PBMC cells. When applied in this setting, we followed the default pipeline recommended by the authors: normalization with SCTransform, followed by TransferAnchors with supervised PCA, TransferData to predict "celltype.l2" labels from the reference to the query, and finally IntegrateEmbeddings and ProjectUMAP to calculate projected embeddings for the query over the space of the reference. To apply Azimuth to a custom reference atlas (in this case, our mouse TIL reference atlas), it was necessary to recalculate the UMAP using the RunUMAP function from Seurat and setting return.model = TRUE, to obtain a usable model for projection. We verified that this step altered only marginally the shape of the UMAP compared to our atlas (which was generated with the "umap" R package). We then applied the same steps outlined above for label transfer and projection, except that we utilized Log-Normalize as a normalization method for query datasets, and set PCA as the reduction method, consistently with the transformations applied to construct the TIL reference atlas. Both for the cross-validation benchmark and the projection of human query data, we transferred the "functional.cluster" labels from the reference to the query using the TransferData function, and compared these predicted cell states with the original annotation.

Scmap[17] is a method that enables mapping cell type identities between experiments, and could be applied to classify query data using the subtypes of our mouse TIL atlas. First, we ran scmapCell using default parameters to find the 10 nearest neighbor reference cells for each query cell. Then, the scmapCell2Cluster function was applied over the nearest neighbors (with parameters threshold = 0 and w = 1) to predict subtypes for each cell in the query dataset.

**T cell classification with singleR.** SingleR[49] is a generic method for single-cell classification that allows annotating cells of a query data set by reference expression profiles. To generate singleR prediction models from human single-cell TIL data from two patient cohorts, we first down-sampled reference sets to a maximum of 1000 cells per original annotation, preserving only annotations represented by at least 100 cells. Then, single-cell profiles for each annotation were aggregated into pseudo-bulk profiles using the singleR "aggregateReference" function, subsetting on genes that were differentially expressed between annotation classes. The functions "trainSingleR" and "classifySingleR" were then applied to generate a predictor and classify cells from a query data set. To train a singleR classifier from human PBMC data, we obtained the normalized counts from the default PBMC Azimuth reference atlas, and calculated pseudo-bulk expression profiles for T cell state annotations as detailed above for the TIL data. To simplify analysis and interpretation, we combined annotations for related T cell subtypes from the reference into a single annotation (CD8 TCM, CD8 Naive, CD4 TCM, CD4 Naive as Naive/TCM; CD4 TEM, CD4 CTL as CD4 TEM/CTL).

**Human-mouse ortholog projection.** We downloaded the list of orthologous genes between human and mouse from Ensemble BioMart release 101. When mapping was ambiguous (i.e., a human gene mapping to several murine genes and viceversa) we favored the identical upper-case human ortholog translation of mouse genes, when available. When projecting human scRNA-seq data onto a mouse reference atlas with ProjecTILs (human.ortho = TRUE), the expression matrix was automatically subset on the human genes with a valid ortholog, and submitted for projection using the mouse gene identifiers. All subsequent analyses were performed in the mouse ortholog space.

Clonal overlap between subtypes was calculated using the Morisita index implementation of scRepertoire[68]. Because of the large imbalance in terms of cell subtypes in the Yost et al. cohort, we uniformly down-sampled all subtypes to a maximum of 500 cells for clonal overlap calculation. For the analysis of gene expression profiles in the Li et al. and Yost et al. cohorts, we calculated the average expression of all cells projected in the CD8_NaiveLike, CD8_EM, CD8_Tpex, and CD8_Tex subtypes for 16 key marker genes. To be able to compare the profile of different genes, we rescaled the average expression value of a given gene by its maximum average expression over the four subtypes under analysis (CD8_NaiveLike, CD8_EM, CD8_Tpex, and CD8_Tex).

**TIL subtype conservation across cohorts, cancer types, and species.** The following datasets were included in the TIL subtype conservation analysis: Azizi et al. breast cancer[4], Carmona et al. melanoma[21], Jerby-Arnon et al. melanoma[69], Li et al. melanoma[5], Nieto et al. colorectal, liver and lung cancer[70], Sade-Feldman et al. melanoma[6], Xiong et al. colon adenocarcinoma[71], Yost et al. basal cell carcinoma[48]. TILs were classified by ProjecTILs using default parameters for murine datasets, and by setting the parameter human = TRUE for human datasets. For each projected dataset, we calculated differentially expressed genes in the

original log-normalized RNA expression space of each dataset, using the FindAllMarkers Seurat function for all reference subtypes represented by at least 50 cells. Genes that were detected as differentially expressed in at least four studies were considered for subsequent analyses, limiting the number of genes per TIL subtype to at most 25 genes. The intersection of differentially expressed genes with variable genes from the reference TIL atlas resulted in 88 genes, from 81 subtype-study combinations, for which we calculated average integrated expression profiles. The resulting data matrix was visualized as a heatmap using the pheatmap package, normalizing data by row (genes), and clustering rows and columns by the ward algorithm.

**Reporting summary.** Further information on research design is available in the Nature Research Reporting Summary linked to this article.

## Data availability

Generated scRNA-seq data of MC38 tumor-draining lymph node T cells were deposited in the ArrayExpress database with accession ID E-MTAB-9274.

To construct the reference TIL atlas, we obtained single-cell gene expression matrices from the following GEO entries: GSE124691[25], GSE116390[21], GSE121478[34], GSE86028[67]; and entry E-MTAB-7919[71] from Array-Express. For the TIL projection examples (OVA Tet+, miR-155 KO and Regnase-KO), we obtained the gene expression counts from entries GSE122713[28], GSE121478[34] and GSE137015[37], respectively.

Single-cell data to build the LCMV-specific CD8+ T cell reference map were downloaded from GEO under the following entries: GSE131535[38], GSE134139[39] and GSE119943[40], selecting only samples in wild type conditions. Data for the Ptpn2-KO, Tox-KO, and CD4-depletion projections were obtained from entries GSE134139[39], GSE119943[40], and GSE137007[45] and were not included in the construction of the reference map. Single-cell expression matrices for LCMV-specific CD8+ T cells in multiple tissues (Fig. 5) were kindly provided by the authors[47]; raw single-cell data are also available at ENA under accession code PRJEB36998.

Processed single-cell RNA-seq gene expression matrices from cancer patient samples were downloaded from GEO under the following entries: GSE123139 (Melanoma_Li)[5], GSE123813 (BasalCC_Yost)[48], GSE120575 (Melanoma_Sade-Feldman)[6], GSE115978 (Melanoma_Jerby-Arnon)[5], GSE114727 (Breast_Azizi)[4]. For the liver, lung, and colorectal cancer samples, we used the single-cell expression matrices collected by Nieto et al.[70] (https://doi.org/10.5281/zenodo.4263972).

Source data for the TIL and viral reference atlases were deposited in figshare with DOI (https://doi.org/10.6084/m9.figshare.12478571)[72] and (https://doi.org/10.6084/m9.figshare.12489518)[73], respectively.

All other data are provided in the article and its Supplementary files or from the corresponding author upon reasonable request. Source data are provided with this paper.

## Code availability

ProjecTILs is freely available as an R package at: https://github.com/carmonalab/ProjecTILs. The package release used for this study is available at DOI[74] (https://doi.org/10.5281/zenodo.4601466).

Several reproducible case studies that apply ProjecTILs for the analysis of published single-cell datasets can be found at: https://github.com/carmonalab/ProjecTILs_CaseStudies.

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

## Acknowledgements
We are grateful to Dr. Amaia Martinez-Usatorre, Prof. Werner Held, and Dr. Gonzalo Parra for critical reading of the paper; to Dr. Ioana Sandu, Prof. Annette Oxenius, and Prof. Manfred Claassen for fruitful discussions on ProjecTILs analysis of LCMV-specific CD8 T cells across tissues. This research was possible thanks to the support of the Swiss National Science Foundation (SNF) Ambizione grant 180010 to SJC. SJC would like to thank the SNF for supporting researchers developing their early independent academic careers in Switzerland, regardless of their nationalities.

## Author contributions
M.A. and S.J.C. designed and developed methods and software, performed computational experiments, interpreted data, and wrote the paper. J.C.O. and G.C. contributed to interpretation of data and manuscript review. S.M. and R.C. performed scRNA-seq experiments and contributed to paper review. S.J.C. conceived and supervised the project. All authors approved the paper.

## Competing interests
MA, JCO, SJC declare no competing interests. GC has received grants, research support or is coinvestigator in clinical trials by BMS, Celgene, Boehringer Ingelheim, Roche, Iovance, and Kite; has received honoraria for consultations or presentations by Roche, Genentech, BMS, AstraZeneca, Sanofi-Aventis, Nextcure, and GeneosTx; he has patents in the domain of antibodies and vaccines targeting the tumor vasculature as well as technologies related to T-cell expansion and engineering for T-cell therapy; and he receives royalties from the University of Pennsylvania for CAR-T technologies licensed to Novartis. SM and RC are employees of Genentech, Inc, a member of the Roche family and receive salary and stock from Roche.
