## [Peer Review File · Nature Communications]

REVIEWER COMMENTS

Reviewer #1 (Remarks to the Author):

In this study by Andreatta et al., the authors developed a new algorithm 'ProjectTILs', which enable the projection of new scRNA-seq data into T cell reference atlases of cancer and viral infection. The algorithm annotates T cells subtypes from new scRNA-seq experiments, but more importantly predicts gene expression changes due to different perturbations. In addition, an important aspect of the algorithm is that it allows comparison between different studies and conditions by projection of different datasets onto the same reference map. The T cell atlases are based on murine single cell datasets, but human T cells data can also be analyzed using orthologous genes between the two species.

With the rapid growth of single cell RNA-seq studies and the current wealth of publicly available single cell data of PBMCs, it will be interesting to expand 'ProjectTILs' atlas to additional immune cell types in order to generate a more complete atlas of the immune system. To date annotation of single cell data is to some extent subjective, and there is lack of standardization, which make it difficult to compare and integrate conclusions from different studies. To this aim expansion of 'ProjectTILs' to the entire repertoire of immune cell types might lay a foundation for standardization in terms of annotation and integration of different studies.

My main concern is that there is still a batch effect in the T cells atlases which can be confounding affect in the annotation process, downstream analysis of the algorithm, and hence the conclusions drawn from the algorithm's results.

Major Comments:

1. According to figure 1B, I'm afraid that there is still a batch effect in the T cells cancer atlas. There are large areas in the UMAP that contain only cells from a specific dataset, e.g. CD8 naïve like cells at the bottom right of the map which are coming almost exclusively from Carmona dataset; moreover in the text the authors state that TILs samples (as the Carmona dataset) are mainly enriched for Tex, T_pex and T_{reg}. However, this sample contains naïve and EM T cells at specific location in the map, maybe this deviation in composition is due to batch effect? Another example is the Megan dLN dataset which is located only in a specific location in the reference map, and its composition is very different from the other dLN sample. In addition, in the method section the author state that after unsupervised clustering and manual annotation of the reference atlas several clusters were merged, I am wondering whether some of these clusters were due to different datasets with the same subtype, this might hint that indeed there is still a batch effect in the atlas after correction. The authors should show this data, e.g. by a stacked barplot with the percentage of cell clustered from each dataset to each cluster (before merging the clusters, with its annotation). Another important factor that should be addressed is the protocol with which each dataset was generated, which also can be a confounding effect, and we need to validate that it is not affecting the reference data.

2. One of the main advantages of the algorithm, as the authors suggest, is that it can predict the effects of perturbation. Indeed, the authors demonstrated that their algorithm can find differentially expressed and regulated genes in different conditions. However, the author should also show that their algorithm is better than the common analysis of single cell data. i.e. the authors should analyze such a dataset alone, perform clustering, classification and differential gene expression, and show that the algorithm's results are superior to the 'regular' analysis.

3. In the manuscript Andreatta et al. classify human TILs onto mouse TILs atlas, and claim that there is a remarkable conservation between human and mouse based on their results. I think the authors should be very cautious with this claim, since projection of human data onto mouse may not be adequate and defining human subtypes based on mouse space may lead to misinterpretation of the data. Therefore, I feel that the author should validate this result experimentally, or use human data as a reference map to show that the results using the mouse atlas are robust and accurate.

Minor comments:

1. When projecting the Xiong et al. dataset (which is part of the reference data) into the reference map, it is not clear whether the same data that is projected is part of the reference data (figure

2D). In addition, it is not clear how this analysis is different from the cross-validation presented in figures S2 and S3.

2. The projection of Miller et al. dataset onto the reference map (figure S4) is the first time the authors present projection of new data, not exist in the reference, onto the reference atlas. It is not clear to me why this section is in the supplement; I would move it to the main text instead of the cross-validation with Xiong et al. dataset.

3. if I understood correctly, the WT data from Ekiz et al. was used to generate the reference dataset. If so, what is the meaning of projecting it onto the reference dataset in figure 3A? The same in figure 4H – the WT data of Lafleur is from the reference dataset. It is very confusing and misleading, especially since the authors claim that it is new datasets (page 13 line 3). In addition, if these are the same cells as the reference, what is the meaning of comparing their gene expression (WT vs. reference in figure 4J).

In general, in many parts along the manuscript it is not clear whether the projection is of the reference dataset or new dataset, the author should clarify it, and explain what is the meaning of it beside cross-validation...

4. what score was used to calculate the association between ICA components and MSigDB signatures?

5. Regarding the classification of human TILs onto mouse TILs atlas - since there are some discrepancies between the original annotations and ProjectTILs annotations, I would suggest to use an independent tool for human single cell annotations (such as singleR) in order to show that the different classifications are not artificial due to projection of human data into mouse reference.

Other comments:

Figure 1: add to the legend of the dataset its origin (e.g. TILs, dLN, CD4, CD8 etc.); relevant also to figure 4B, add the experimental condition of each dataset (e.g. acute/chronic infection 4.5/7.5 days).

Figure 1E-F: there is no need to show the gene expression both in violin plot and on the UMAP.

Figure 2E: add to the legend explanation about this plot, is it normalized expression? Z-score?

Figure 3C: I think there is a confusion in the legend between the WT and KO cells, otherwise it is not clear, the green points (KO) seems more cycling, while in the text it is written that "The KO TILs also scored lower in terms of cycling signature".

Reviewer #2 (Remarks to the Author):

The manuscript, "Projecting single-cell transcriptomics data onto a reference T cell atlas to interpret immune responses", by Andreatta and colleagues, describes a novel algorithm ProjectTILs, which enables the projection of scRNA-seq data onto reference atlases to systematically interpret cellular states across studies and diseases. They used ProjectTILs to conduct the analysis on both mouse and human tumor models to explore the effects of multiple perturbations, including immunoregulatory target genes as well as clonotype structure, which supply a new approach for single-cell classification and identification. However, aside from the algorithm of this study, most of the key observations just confirm things that were previously known. There are some limitations and concerns as detailed below.

1. One of the greatest strengths of single cell studies is the ability to explore rare and potentially important cells, which are easily masked by other bystander cells during bulk sequencing. The authors projected all cells to the reference atlases containing limited number of cell subtypes. Even though it's easier to define most cells' identities, rare cells with low cell numbers will likely not be detected under this strategy and these cells may be very important to reflect the subtle changes during immunotherapy treatment. In addition, rare cells are more likely linked to a specific tumor microenvironment, such as CXCL13+BHLHE40+ TH1-like cells identified in the study "Lineage tracking reveals dynamic relationships of T cells in colorectal cancer", which were found to be preferentially enriched in patients with microsatellite-unstable tumors. Thus a major concern of ProjectTILs is the limited reference sets/cells used in their atlas. The authors should test whether ProjectTILs can detect/classify rare subsets from TILs derived from various human cancers.

2. Even though the authors successfully projected TIL from either basal cell carcinoma cohort by Yost et al. or melanoma cohort by Li et al., they didn't provide data on how well the results from other models, such as the mouse models as well as miR-155 KO and Regnase-1 KO models, projected onto their Atlas. Are there any big differences in their projected classifications compared to the classifications derived from these original studies?

3. The authors incorrectly used the term "tumor reactive T cells" in the manuscript. Although these cells express some hallmarks/genotypic markers of tumor-reactive T cells, they were not formally tested and found to be tumor-reactive. Thus, those cells should be categorized to tumor specific T cells but not tumor reactive T cells.

Reviewer #3 (Remarks to the Author):

The authors propose a novel method (named ProjectTILs) that projects new scRNA-seq data onto the reference atlases for cancer and viral infection. ProjectTILs predicts the effects of multiple perturbations and conducts a meta-analysis by projecting human tumor-infiltrating T lymphocytes (TILs) onto a mouse atlas. Below are a few major concerns:

1. The authors should perform better literature research: for example, many methods have been developed to map the single cells to reference atlases. The novelty and the significance of developing ProjectTILs is not clear in the manuscript.
2. In the benchmark part, the authors only show the robustness of the method by cross-validation experiments. The authors should compare with the other methods, including but not limited to scmap.
3. scRNA-seq data may contains unknown cell types that are not reported by the references. How is the performance of ProjectTILs in this situation? The authors should also compare the results with the other software, such as MARS.
4. In Figure 2, the authors project a scRNA-seq dataset onto the reference TIL atlas, a dataset of tumor-infiltrating T cells. The authors only analyze the component of subclones in each cluster without explaining more detailed results about the association between clusters and subclones. The authors may find another time-course dataset to show the relationships. Furthermore, Figure 2G is unclear, such as the definition of the level of clonal expansion.
5. ProjectTILs uses PCA or ICA for dimension reduction. Please set some criteria to evaluate the performance in different data scenarios.
6. Please use the consistent wording in the paper: for example, CD8 EM, CD8_EM, CD8_EffectorMemory may refer the same cluster.

Response to the reviewers

Manuscript **NCOMMS-20-40115-T** - "Projecting single-cell transcriptomics data onto a reference T cell atlas to interpret immune responses" by Andreatta et al.

We thank the reviewers for their time and their comments. In the following letter, we address their concerns point by point, and detail new analyses and updates to the manuscript.

Briefly, we have addressed the following major points:

- We performed new analyses to address the potential residual batch effects in the TIL atlas (Supplementary Fig. 1 and Auxiliary Figure 1), and added a new table (Supplementary Table 1) to indicate the T cell populations comprised by each study.
- To compare the proposed analysis pipeline to "standard" single-cell analysis protocols (i.e. clustering, annotation, differential expression), we applied our method to a very recent dataset of CD8 T cells in infection across multiple tissues, and demonstrated that we could obtain very similar results to the original study in an automated fashion and without requiring expertise about the specific system (new Figure 5). This analysis also addresses the concern by reviewer #2 as to the ability of ProjecTILs to recapitulate the original cell annotations from mouse studies.
- As suggested by reviewer #3, we included performance comparisons to other methods (Azimuth and scmap) and updated the text to better formulate how the proposed method is unique compared to available single-cell analysis tools; as suggested by reviewer #1, we also applied singleR to resolve annotation discrepancies for human T cell data. Several new supplementary figures document these analyses (Supplementary Fig. 3, 10, 11, 12 and 13).
- To address the concerns of T cell state conservation between mouse and human, we extended our analysis from 2 patient cohorts/cancer types to 8 patient cohorts, encompassing 132 tumor samples and 7 different cancer types. We demonstrated that T cells from different species and cancer types displayed conserved transcriptional profiles, clustering by TIL subtype rather than by sample of origin, tissue or species (new Figure 7). In Supplementary Data 1, we provide marker genes, as well as full expression profiles, for the reference TIL subtypes in mouse and human.
- Several other more specific concerns are addressed in detail below. Reviewer comments are reported in *italics* and quotes from the revised manuscript in red.

REVIEWER COMMENTS

Reviewer #1 (Remarks to the Author):

In this study by Andreatta et al., the authors developed a new algorithm 'ProjecTILs', which enable the projection of new scRNA-seq data into T cell reference atlases of cancer and viral infection. The algorithm annotates T cells subtypes from new scRNA-seq experiments, but more importantly predicts gene expression changes due to different perturbations. In addition, an important aspect of the algorithm is that it allows comparison between different studies and conditions by projection of different datasets onto the same reference map. The T cell atlases are based on murine single cell datasets, but human T cells data can also be analyzed using orthologous genes between the two species.

With the rapid growth of single cell RNA-seq studies and the current wealth of publicly available single cell data of PBMCs, it will be interesting to expand 'ProjecTILs' atlas to additional immune cell types in order to generate a more complete atlas of the immune system. To date annotation of single cell data is to some extent subjective, and there is lack of standardization, which make it difficult to compare and integrate conclusions from different studies. To this aim expansion of 'ProjecTILs' to the entire repertoire of immune cell types might lay a foundation for standardization in terms of annotation and integration of different studies.

My main concern is that there is still a batch effect in the T cells atlases which can be confounding affect in the annotation process, downstream analysis of the algorithm, and hence the conclusions drawn from the algorithm's results.

Major Comments:

1. According to figure 1B, I'm afraid that there is still a batch effect in the T cells cancer atlas. There are large areas in the UMAP that contain only cells from a specific dataset, e.g. CD8 naïve like cells at the bottom right of the map which are coming almost exclusively from Carmona dataset; moreover in the text the authors state that TILs samples (as the Carmona dataset) are mainly enriched for Tex, T_{pex} and T_{reg}. However, this sample contains naïve and EM T cells at specific location in the map, maybe this deviation in composition is due to batch effect? Another example is the Megan dLN dataset which is located only in a specific location in the reference map, and its composition is very different from the other dLN sample. In addition, in the method section the author state that after unsupervised clustering and manual annotation of the reference atlas several clusters were merged, I am wondering whether some of these clusters were due to different datasets with the same subtype, this might hint that indeed there is still a batch effect in the atlas after correction. The authors should show this data, e.g. by a stacked barplot with the percentage of cell clustered from each dataset to each cluster (before merging the clusters, with its annotation). Another important factor that should be addressed is the protocol with which each dataset was generated, which also can be a confounding effect, and we need to validate that it is not affecting the reference data.

Response - We thank the reviewer for bringing attention to the importance of comprehensively addressing batch effects when constructing cell atlases. In the revised manuscript, we present new evidence and perform additional analyses to demonstrate the absence of strong batch effects in our atlases.

With specific regard to Figure 1B mentioned by the reviewer, we believe that two main confounding factors may explain the impression of strong batch effects: i) different studies/dataset are composed of different subtypes (e.g. total TIL, CD8+ TIL, PD1+ CD4+ TIL); and ii) the type of visualization of Figure 1B, where cells are overlaid and larger datasets mask smaller ones, may amplify this discrepancy. We have now included a new table (Supplementary Table 1) indicating the type of isolated T cell population in each study and the sequencing protocol/technology used. Next, for each of the studies we have included an unsupervised analysis followed by manual cluster annotation, indicating the original subtype composition (irrespective of the TIL atlas, but using the same subtype nomenclature) and cell location on the reference atlas (new Supplementary Fig. 1). This updated figure highlights the great diversity of subtype composition among different studies. For instance, while follicular helper T cells (T_{fh}) constitute about 2% of the MC38_dLN sample (containing both CD8 and CD4 T cells), they account for more than 90% of the Magen_dLN sample (composed of tumor-specific CD4 T cells only). These differences in cell type composition explain the discrepancy noted by the reviewer about these dLN samples covering different areas of the UMAP.

We have updated the manuscript to highlight the agreement between unsupervised subtype definition in each dataset with atlas annotation:

From line 117

We confirmed that clusters identified by unsupervised analysis of individual datasets were largely consistent with corresponding “functional clusters” shared by multiple datasets in the integrated reference atlas (Supplementary Fig. 1).

For the benefit of this reviewer, we also provide below more detailed visualizations – split by datasets and subtypes – demonstrating that most of the UMAP areas are covered by all relevant studies, and that all UMAP areas are covered by at least two studies (Auxiliary Figure 1).

Supplementary Fig. 1: Unsupervised clustering, annotation and composition of datasets included in the reference TIL atlas.

Auxiliary Figure 1: distribution of cells on the TIL reference atlas by sample/dataset (columns) and subtype (rows).

The reviewer further mentioned that: "...this sample (Carmona) contains naïve and EM T cells at specific location in the map". Indeed, a closer examination of Auxiliary Figure 1 shows that EM TILs from the Carmona dataset are preferentially located to the left side of the CD8_EM cluster, while EM cells from the Xiong and Singer datasets are preferentially located on the right area of the CD8_EM cluster. However, EM cells from the Ekiz study are well distributed across all the CD8_EM cluster. Hence, these intra-subtype location biases per study are more likely to reflect true biological heterogeneity within the EM subtype rather than batch effects (e.g. heterogeneous expression of *Gzmb* and *Cx3cr1*, Figure 1 F). Moreover, as indicated in the new Supplementary Table 1, these differences are not explained by scRNA-seq technology, as Singer et al. used smart-seq2 while the Xiong, Carmona and Ekiz dataset were generated with the 10x technology.

If we examine the naive-like subtype, as the reviewer noted, the MC38_dLN and the Carmona datasets preferentially occupy different regions of the naive-like cluster. However, these regions are also covered by the rest of the studies (Ekiz, Singer and Xiong) and therefore are also more likely to reflect true biological differences, such as different composition of naive and central memory cells, tissue-specific differences, etc. Indeed, the naive-like subtype may contain both bona fide naive cells as well as central memory cells (hence the name "naive-like"). We have updated the manuscript to highlight this aspect (line 104).

Finally, it would certainly be possible to increase clustering resolution to obtain a more granular definition of subtypes. In this study, and at this stage, we have chosen to define broad TIL subtypes that cover the main T cell subtypes in infection and cancer. As more data become available, we will be able to confidently increase resolution of our atlases and define more specific subtypes.

2. One of the main advantages of the algorithm, as the authors suggest, is that it can predict the effects of perturbation. Indeed, the authors demonstrated that their algorithm can find differentially expressed and regulated genes in different conditions. However, the author should also show that their algorithm is better than the common analysis of single cell data. i.e. the authors should analyze such a dataset alone, perform clustering, classification and differential gene expression, and show that the algorithm's results are superior to the 'regular' analysis.

Response - We do not claim that results obtained using ProjecTILs cannot be obtained using other combinations of tools. On the contrary, we believe that, in general, conclusions derived from scRNA-seq analysis should be supported by multiple methodological approaches. The crucial advantage of ProjecTILs is that, instead of requiring weeks to months of work by expert bioinformaticians in collaboration with immunologists to properly discriminate batch from biological effects and correctly annotate T cell states by extensive literature curation and gene signature comparison, ProjecTILs can do an equally good job in minutes in an automated way.

To emphasize this aspect, we have included a new analysis in the revised manuscript, where we apply ProjecTILs to a recent dataset of mouse CD8 T cells in chronic infection (Sandu et al. Cell Reports, August 2020). In this work, the authors set out to describe the heterogeneity of CD8 T cells across multiple tissues, using a "standard" approach based on dimensionality reduction, unsupervised clustering, differential expression analysis and manual cluster annotation. In their paper, the definition of cell clusters, including considerations about batch-effects vs. tissue-related biological differences, the annotation of meaningful cell types, the differential expression analyses to determine inter-subtype differences as well as inter-tissue differences, all required an enormous amount of curation and expertise about the system under study. In the revised manuscript we show how our analysis pipeline can lead to very similar results with minimal effort and domain expertise (new Figure 5).

Although our atlas was constructed using spleen-derived samples only, we found that tissue-specific composition of T cell subtypes predicted by ProjecTILs strongly correlates with the subtypes defined by the original study in an unsupervised way (Fig 5 B-D). Moreover, ProjecTILs detected specific genes and gene modules associated with specific tissues and T cell subtypes (Fig 5 E-G), for example, lower activation-related genes in effector cells in blood vs. spleen (*Nfkb1a*, *Nr4a1*, *Cd69*), in full agreement with the conclusions of the original study. The results are reproducible with the code posted as a case study at https://carmonalab.github.io/ProjecTILs_CaseStudies/Sandu_LCMV.html, highlighting how meaningful conclusions can be seamlessly drawn using the ProjecTILs pipeline.

Figure 5: ProjectTILs resolves tissue-specific T cell heterogeneity in chronic viral infection

The manuscript was updated to include this analysis, from line 282:

A crucial variable affecting T cell heterogeneity is their environment. A recent study by Sandu *et al.* investigated the diversity of CD8 T cells in chronic LCMV infection across six different tissues, and determined organ-specific transcriptomic profiles that could be divided into five main functional subtypes. Taking advantage of our reference CD8 T cell atlas for viral infection, we re-analyzed the data from Sandu *et al.* to determine if tissue-specific transcriptomic alterations could be detected using our automated ProjectTILs pipeline (Figure 5 A). In general, the majority of virus-specific T cells were predicted to be terminally exhausted (Tex) as expected in this infection model, but different tissues were composed of variable fractions of other T cell subtypes (Figure 5 B). For instance, lung, blood and spleen had the highest percentage of effector cells (SLEC), while lymph node and spleen had an exceeding percentage of Tpex cells compared to other tissues. While the original study defined fewer T cell states compared to our reference atlas for infection (Figure 5 C), the tissue-specific composition for the main T cell subtypes showed a remarkable correspondence between the ProjectTILs prediction and the original, unsupervised analysis (Figure 5 D). A unique advantage of projection into a stable reference atlas is that multiple samples (or multiple tissues, in this case) can be compared over the same reference, and for specific T cell subtypes. Differential expression between SLEC from blood and

spleen showed that SLEC in spleen overexpress markers of activation such as *Nfkbia*, *Nr4a1* and *Cd69* (Figure 5 E), indicating that these cells may have recently encountered antigen, unlike circulating cells. A similar observation can be made by comparing Tex cells from liver and spleen, but in this case also a significant overexpression of *Gzma* in liver is observed (Figure 5 F), as also noted in the original study. Interestingly, two of the most discriminant ICA components between spleen and other tissues contain several genes involved in T cell activation and TCR signaling (Figure 5 G). This analysis demonstrates that, given an appropriate reference atlas, ProjecTILs can detect tissue-specific signals using a fully automated pipeline and default parameters, and obtain very similar results compared to a manually curated analysis performed by expert immunologists and bioinformaticians.

In the updated manuscript, we also included a comparison of ProjecTILs analysis to other methods for cell annotation and projection (Azimuth and scmap), and showed that ProjecTILs outperforms the others in a cross-validation experiment over the TIL mouse reference atlas (Supplementary Fig. 3), as well as for the projection and annotation of human T cell data (Supplementary Fig. 13).

From line 145:

Benchmarking ProjecTILs by cross-validation experiments showed a high accuracy (>90%) both for the projection (Supplementary Fig. 2, and Methods) and classification tasks, significantly outperforming Azimuth and scmap (Supplementary Fig. 3).

From line 332:

As an alternative projection algorithm to interpret human TIL states, we applied Azimuth, a recently developed method that by default utilizes a human PBMC reference atlas. We applied Azimuth to project T cell data from the Yost *et al.* cohort using the PBMC reference provided by the authors, as well as using our mouse TIL atlas. While broad T cell states (CD4+ vs. CD8+, Tregs) could be distinguished in the projections on the PBMC atlas, Azimuth could not discern between more specific subtypes. A considerable fraction of T cells were also assigned to the NK and MAIT areas of the reference map (Supplementary Fig. 12). When Azimuth was applied to the more specialized TIL reference atlas developed by us, projections appeared reasonable for certain cell types (Treg, Tfh, CD8 effector), but naive-like cells were mostly misclassified, as well as exhausted T cells (Supplementary Fig. 13 A-B). On a subset of cell subtypes that could be confidently mapped between studies, Azimuth resulted less accurate than ProjecTILs for the classification task (Supplementary Fig. 13 C-D). These results highlight the importance of an accurate projection algorithm, but also of a robust atlas, specific for the problem at hand, rather than generalist, whole tissue atlases.

3. In the manuscript Andreatta et al. classify human TILs onto mouse TILs atlas, and claim that there is a remarkable conservation between human and mouse based on their results. I think the authors should be very cautious with this claim, since projection of human data onto mouse may not be adequate and defining human subtypes based on mouse space may lead to misinterpretation of the data. Therefore, I feel that the author should validate this result experimentally, or use human data as a reference map to show that the results using the mouse atlas are robust and accurate.

Response - Our claim about the remarkable conservation between human and murine TIL subtypes was based on the observation that human TIL diversity from two cohorts/cancer types, as defined in those studies, was very well explained by the projection of these cells on the murine TIL atlas, with clear discrimination of dysfunctional/exhausted, pre-dysfunctional, memory, cytotoxic/effector CD8 T cell, Tregs, CD4 T helpers and naive cells.

In this revised version, we provide new and strong evidence to support our claim. We extended our analysis from two to 8 patient cohorts (+2 murine studies), projecting 106,103 TILs from 132 tumor samples, covering 7 different cancer types. We found that average expression profiles from murine and human samples clustered by TIL subtype and not by than by sample of origin, tissue or cohort/study (new Figure 7). We included a probabilistic analysis to show that the chances of explaining this pattern by chance are extremely low (p -value $< 3 \times 10^{-6}$).

Figure 7: Conservation of T cell subtypes across studies, cancer types and species

Importantly, major known T cell type markers (as well as potentially novel candidates) were consistently differentially expressed in their respective reference subtypes, regardless of the cancer type or species. In the updated manuscript, we are also making available the marker genes identified by this meta-analysis, as well as complete average expression profiles for each TIL reference subtype, both for mouse and human (new Supplementary Data 1).

Note that we do not claim that T cell subsets which are specific to humans do not exist, but rather that the nine major TIL subtypes defined in our reference atlas are highly conserved between the two species. While human-specific TIL subsets are likely to exist, they might be restricted to low-frequency populations or to certain tissues or conditions. Instead, our results indicate that major TIL subtypes in our reference TIL atlas explain most of murine and human TIL diversity, for the cancer types and tissues considered in the analysis.

New text in the manuscript (from line 345):

To further study the conservation of the reference TIL subtypes of our atlas, we analyzed scRNA-seq data from 132 tumor biopsies from 10 different studies, covering 7 different cancer types. After applying ProjecTILs projection and classification, we determined the differentially expressed genes of each TIL subtype across all datasets (see Methods), and used these to calculate average expression profiles for individual studies and TIL subtypes (Figure 7, Suppl. Data S1). For example, *Gzma*, *Gzmk* and *Ccl5* were significantly more expressed in CD8 EM cells from most cancer types, both murine and human; *Pdcd1*, *Havcr2* and *Prf1*, among other genes, identified CD8 exhausted cells; *Xcl1*, *Tnfrsf4*, and *Tox*, among others, marked precursor exhausted CD8 T cells; *Sell*, *Tcf7*, *Il7r* and *Ccr7* were enriched in both

CD4 and CD8 naive-like cells; *Foxp3*, *Il2ra*, *Ctla4* and several other genes identified Tregs; *Tox2* and *Tbc1d4* were differentially expressed in Tfh; and *Cd40lg*, *Anxa1* and *Rora*, were enriched in T helper cells. Multiple other genes without previously documented associations with tumor-infiltrating T cells were also identified, revealing interesting targets for future validation (Supplementary Data 1). Overall, average expression profiles clustered preferentially by TIL reference subtype rather than by study, cancer type or species (Figure 7). In particular, observing human and murine samples clustered together in each of the nine reference subtypes is statistically significant (p -value $< 3 \times 10^{-6}$), and points to a large conservation between human and mouse TIL states.

Minor comments:

1. When projecting the Xiong et al. dataset (which is part of the reference data) into the reference map, it is not clear whether the same data that is projected is part of the reference data (figure 2D). In addition, it is not clear how this analysis is different from the cross-validation presented in figures S2 and S3.

Response - The reviewer is correct in that the data being projected is already present in the reference map. The data presented in Figure 2 were meant to be an illustrative example of projection, to show schematically the steps of the projection algorithm, and the kinds of analysis it enables. We agree that it may be confusing to re-project data already present in the reference as the first example of projection, and have now replaced panel 2D-G with the projection of the data from Miller et al. (as suggested in the following point by this reviewer).

2. The projection of Miller et al. dataset onto the reference map (figure S4) is the first time the authors present projection of new data, not exist in the reference, onto the reference atlas. It is not clear to me why this section is in the supplement; I would move it to the main text instead of the cross-validation with Xiong et al. dataset.

Response - We thank the reviewer for the suggestion, we have now moved this analysis from Figure S4 to the main Figure 2 (see also previous response).

3. If I understood correctly, the WT data from Ekiz et al. was used to generate the reference dataset. If so, what is the meaning of projecting it onto the reference dataset in figure 3A?

The same in figure 4H – the WT data of Lafleur is from the reference dataset. It is very confusing and misleading, especially since the authors claim that it is new datasets (page 13 line 3). In addition, if these are the same cells as the reference, what is the meaning of comparing their gene expression (WT vs. reference in figure 4J).

In general, in many parts along the manuscript it is not clear whether the projection is of the reference dataset or new dataset, the author should clarify it, and explain what is the meaning of it beside cross-validation...

Response - Indeed, in the case of the Ekiz et al. data and the Lafleur et al. data, the WT samples were included in the reference atlases. However, the two studies also included interesting genetic alterations (miR155-KO and Ptpn2-KO respectively) that could be studied using ProjectTILs and that were not part of the reference. While it is a relatively easy task to project the WT samples (as they are already contained in the reference), it is also not completely trivial. This is because the reference is constructed from multiple datasets, reducing batch effects between samples, and constructing a common space that is determined by these multiple datasets. Therefore, re-projection of a dataset already in the reference is performed against a larger and more complex space that is only in part determined by the dataset in question.

For these reasons, we subjected both the WT and KO samples to the same projection pipeline. This ensures that paired conditions are processed uniformly and that they can be compared over the same reference atlas. In the same way, comparing gene expression is still relevant to observe differences in specific genes between a WT and its KO. In the example mentioned by the reviewer (Fig 4J), the gene profile specific to the WT query set will in general be different from the gene profile of the reference, which is determined over the multiple datasets contained in the reference.

Therefore, re-projection of paired conditions into the reference, even in cases when one is already present in the reference, appears to be appropriate. We have updated the manuscript to clarify this point.

From line 166:

Note that, although the WT sample from this study was included in the construction of the reference map, for this analysis it was re-projected using the same pipeline applied to the *miR-155* KO sample. This ensures that the two conditions were processed and projected uniformly and that they could be compared over the same reference map.

4. what score was used to calculate the association between ICA components and MSigDB signatures?

Response - We scored each ICA against these signatures by summing the ICA gene loadings for all genes in a given signature, and then taking the absolute value of this score. In the revised manuscript we added this information in the Methods, section “ICA and discriminant dimensions” (line 633):

5. Regarding the classification of human TILs onto mouse TILs atlas - since there are some discrepancies between the original annotations and ProjectTILs annotations, I would suggest to use an independent tool for human single cell annotations (such as singleR) in order to show that the different classifications are not artificial due to projection of human data into mouse reference.

Response - In perhaps the most notable case of discrepancy between the original annotation and ProjectTILs classification, we saw that a significant fraction of the cells annotated by the authors as “exhausted/dysfunctional” were projected on the effector-memory state of the murine TIL atlas. By examining the expression profile of these cells, we could confirm that these cells did indeed display a clear effector-memory gene profile (i.e. high *GZMK*, *GZMA* and *GZMB* expression) but lacked markers of exhaustion, such as *TOX*, *ENTPD1*, *HAVCR*, *PDCD1* (Figure 5C). This indicates that a fraction of exhausted/dysfunctional cells did in fact better correspond to an effector-memory expression profile.

To better substantiate this observation, we followed the suggestion of the reviewer and applied singleR to annotate TILs from the two patient cohorts. We trained singleR predictors from the expression profiles and annotations of one cohort, and used these models to annotate cells of the second cohort. In particular, we focused on cases where there were major discrepancies between the original annotations and ProjectTILs annotations (dysfunctional/exhausted cells, and naive-like cells). This experiment confirmed that cells originally annotated as dysfunctional/exhausted were in fact, composed of a mixture of exhausted and cytotoxic/effector-memory cells, both according to the singleR predictor and ProjectTILs annotations (Supplementary Fig. 11 A-B). In a similar way, tumor-infiltrating human T cells originally annotated as “naive” appeared to be heterogeneous and could not be unequivocally classified as naive by the singleR predictor trained on the other cohort (Supplementary Fig. 11 C-D). As a further independent classifier, we trained a singleR model from human PBMC data from Hao et al., 2020 (<https://doi.org/10.1101/2020.10.12.335331>) which should be well represented in terms of bona fide naive cells. For both cohorts, the singleR model trained on PBMC profiles suggested that cells originally annotated as “naive” were in fact a mixture of at least two T cell subtypes: naive/central-memory cells and CD4 effector-memory cells (Supplementary Fig. 11 E-F). These results are in line with ProjectTILs classifications, which predicted that “naive” cells (according to the original annotation) were mainly a combination of naive-like and T helper cells.

Other comments:

Figure 1: add to the legend of the dataset its origin (e.g. TILs, dLN, CD4, CD8 etc.); relevant also to figure 4B, add the experimental condition of each dataset (e.g. acute/chronic infection 4.5/7.5 days).

Response - We added a new supplementary table (Supplementary Table 1), that describes the type of isolated T cell population in each study, among other information. We also included in the legend for Figure 4 a brief description of the day/type of infection for each batch in Figure 4B (line 991).

Figure 1E-F: there is no need to show the gene expression both in violin plot and on the UMAP.

Response - We believe these two pieces of information are not redundant. Violin plots are good for displaying the average expression of a given gene for a defined cell subtype, whereas the UMAP representation can highlight gradients of expression within and across clusters/subtypes (e.g. the gradient of expression of *Cx3cr1* in the T_{pex}/T_{ex} states, or the heterogeneous *Gzmb* and *Gzmk* expression within the effector-memory subset).

Figure 2E: add to the legend explanation about this plot, is it normalized expression? Z-score?

Response - In the radar plots, average gene expression is normalized between 0 and 1. We have updated Figure 2, panels D-G according to minor comment 1 and 2, and added information about the radar plots in the legend.

Figure 3C: I think there is a confusion in the legend between the WT and KO cells, otherwise it is not clear, the green points (KO) seems more cycling, while in the text it is written that “The KO TILs also scored lower in terms of cycling signature”.

Response - Thank you for spotting this. Indeed, the labels for WT and KO in the legend were accidentally swapped. We have updated Figure 3 with the correct legend.

Reviewer #2 (Remarks to the Author):

The manuscript, “Projecting single-cell transcriptomics data onto a reference T cell atlas to interpret immune responses”, by Andreatta and colleagues, describes a novel algorithm ProjectTILs, which enables the projection of scRNA-seq data onto reference atlases to systematically interpret cellular states across studies and diseases. They used ProjectTILs to conduct the analysis on both mouse and human tumor models to explore the effects of multiple perturbations, including immunoregulatory target genes as well as clonotype structure, which supply a new approach for single-cell classification and identification. However, aside from the algorithm of this study, most of the key observations just confirm things that were previously known. There are some limitations and concerns as detailed below.

1. One of the greatest strengths of single cell studies is the ability to explore rare and potentially important cells, which are easily masked by other bystander cells during bulk sequencing. The authors projected all cells to the reference atlases containing limited number of cell subtypes. Even though it's easier to define most cells' identities, rare cells with low cell numbers will likely not be detected under this strategy and these cells may be very important to reflect the subtle changes during immunotherapy treatment. In addition, rare cells are more likely linked to a specific tumor microenvironment, such as CXCL13+BHLHE40+ TH1-like cells identified in the study “Lineage tracking reveals dynamic relationships of T cells in colorectal cancer”, which were found to be preferentially enriched in patients with microsatellite-instable tumors. Thus a major concern of ProjectTILs is the limited reference sets/cells used in their atlas. The authors should test whether ProjectTILs can detect/classify rare subsets from TILs derived from various human cancers.

Response - We agree with the reviewer that a big strength of single-cell studies is their ability to detect novel cell populations even when they are present with low cell numbers/frequency (i.e. rare novel populations). Indeed, detection of novel T cell states that might be associated to specific conditions and deviate from the reference (typical) states has been a design principle for our algorithm. Unlike other reference projection algorithms, ProjectTILs provides the means to detect and characterize such novel states. For instance, we have shown that novel states such as the terminally exhausted CD8 TIL state induced by ablation of Regnase-1 – characterized by down-regulation of an inhibitory gene module centered on *Klrc1* and *Lag3* – can be detected even though this state was ‘unknown’ to the reference map (Figure 3).

In this revision, we included a new analysis to demonstrate how ProjectTILs detected tissue-specific states from a recent paper, that were unknown to the reference (see answer to the following comment). As suggested by this reviewer, we also analyzed single-cell data from the paper “Lineage tracking reveals dynamic relationships of T cells in colorectal cancer” (Zhang et al., Nature, 2018). ProjectTILs confirmed that, despite similar TIL subtype composition between microsatellite stable (MSS) and microsatellite-instable (MSI) tumors, Th1 TILs from MSI tumors upregulated *CXCL13* compared to Th1 TILs both from MSS human colorectal tumors and from our murine reference (Auxiliary Figure 2 below). In fact, *CXCL13* was the most differentially expressed gene between MSI and MSS Th1 cells. Of note, *BHLHE40* was high in all Th1 (MSS and MSI irrespectively), and therefore it does not appear to be dependent on tumor genomic stability. This example further confirms the ability of ProjectTILs to detect T cell states that were unknown to the reference.

Auxiliary Figure 2: analysis of MSI vs MSS single-cell data from Zhang et al.

Beyond these examples, the fact that a cell subtype is typically present at low frequencies, is not *per se* a limitation for its inclusion in the reference atlas. On the contrary, our approach to reference atlas construction – based on multi-study integration – allows us to increase the resolution of any part of the reference atlas by including datasets that focus on specific tissues/conditions, as we have done, for instance, with CD4⁺ draining lymph nodes samples to increase resolution of T follicular helper cells, that otherwise would be too ‘rare’ in tumors. Importantly, while detection of a rare population by clustering or unsupervised techniques require a minimum number of cells, ProjectTILs, as a supervised classification method, will correctly project cells on the reference map irrespectively of their absolute

number (e.g. in new Supplementary Figure 1, we see projected cells at any frequencies and cell numbers).

More generally, we believe that a lot of caution should be exercised when claiming the discovery of novel cell subtypes from scRNA-seq data analysis of single cohorts and sometimes even single subjects. The 'rare cell type' detection power is limited by cell multiplet rates. The majority of scRNA-seq data is generated using droplet-based scRNA-seq platforms, where the frequency of multiplets increases with greater loading of cells. Therefore, there is a trade-off between the number of cells that can be studied per sample (high number needed to detect low-frequency cells) and the probability of capturing and sequencing two or more cells together, leading to critical confounding factors. Typical unresolved multiplet rate is 3% and up to 8% using recommended cell dilutions and settings (<https://doi.org/10.1016/j.celrep.2019.09.082>, <https://genomebiology.biomedcentral.com/articles/10.1186/s13059-018-1603-1>). This means that with platforms such as 10X, interpretation of apparently novel populations detected at frequencies below 3% should be evaluated very cautiously, especially when they do not correspond to robustly defined subtypes. For instance, analysis of scRNA-seq datasets of FACS-sorted T cells almost always requires removal of T cell-myeloid cell doublets. For these reasons, ProjecTILs includes a filtering step that removes transcriptomes with non-T cell type signatures, in order to allow only pure T cell transcriptomes to be projected on reference T cell atlas. It is our opinion, and a design principle for our algorithm, that interpretation of single-cell data should start from references of robustly defined, broad subtypes, and only as a second step by characterizing how novel/altered states deviate from the reference subtypes.

2. Even though the authors successfully projected TIL from either basal cell carcinoma cohort by Yost et al. or melanoma cohort by Li et al., they didn't provide data on how well the results from other models, such as the mouse models as well as miR-155 KO and Regnase-1 KO models, projected onto their Atlas. Are there any big differences in their projected classifications compared to the classifications derived from these original studies?

Response - Unfortunately, the studies describing the miR-155 KO and Regnase-1 KO models do not provide annotations for T cells states based on single-cell data, so we cannot compare them with our classification. However, we have included in the revised manuscript the analysis of a very recent dataset of mouse CD8 T cells in chronic infection (Sandu et al. Cell Reports, 2020), which describes the transcriptomics heterogeneity of CD8 T cells across multiple tissues (new Figure 5). We found that the T cell subtypes predicted by ProjecTILs across different tissues correlated very well with the subtypes defined by the original study using unsupervised analysis (Figure 5 B-D). Moreover, we could detect specific genes and gene modules associated with specific tissues and T cell subtypes, for example increased activation-related genes in spleen compared to other tissues (*Nfkb1a*, *Nr4a1*, *Cd69*) (Fig 5 E-G), in line with the conclusions of the original study. These results confirmed both the definition of cell states in our reference atlas, as well as the robustness of the projection method across multiple tissues in mouse models.

This new analysis is described in the revised manuscript, from line 282.

3. The authors incorrectly used the term "tumor reactive T cells" in the manuscript. Although these cells express some hallmarks/genotypic markers of tumor-reactive T cells, they were not formally tested and found to be tumor-reactive. Thus, those cells should be categorized to tumor specific T cells but not tumor reactive T cells.

Response - Thank you for pointing this out. We have updated the manuscript to mark this distinction.

Reviewer #3 (Remarks to the Author):

The authors propose a novel method (named ProjecTILs) that projects new scRNA-seq data onto the reference atlases for cancer and viral infection. ProjecTILs predicts the effects of multiple perturbations and conducts a meta-analysis by projecting human tumor-infiltrating T lymphocytes (TILs) onto a mouse atlas. Below are a few major concerns:

1. The authors should perform better literature research: for example, many methods have been developed to map the single cells to reference atlases. The novelty and the significance of developing ProjectTILs is not clear in the manuscript.

Response - Perhaps we have not emphasized enough how ProjectTILs is unique compared to other available tools for single-cell data analysis. As the reviewer correctly points out, there are several recent methods that allow “mapping single cells to reference atlases”, including scMap, Seurat (in label transfer mode), CellBlast (July 2020), ELSA (June 2020). However, all of these methods start from the premise of defining a new embedding space specific for the query data set, and then use the reference labels to annotate the cells in the query. In contrast, our method focuses on maintaining the reference embedding space intact, and analyses new data in the context of this unaltered reference space. We showed with multiple examples how this allows us to easily compare different conditions, tissues and therapies over the same reference space; a task that would be unwieldy if a new, different embedding space were defined for each new query dataset (i.e. condition, tissue) under analysis.

We further show that a stable, annotated reference atlas can provide additional axes of variation besides the simplistic UMAP representation, and that by definition such axes are the same for the reference and any query, projected dataset; therefore, multiple experimental conditions can be also compared in terms of these additional dimensions, allowing the detection of novel or altered states. In the revised manuscript, we include a new analysis of CD8 T cells in chronic infection that exemplifies the application of ProjectTILs to study the transcriptomics heterogeneity of CD8 T cells across multiple tissues (Figure 5). Not only ProjectTILs recapitulates the fine T cell subtype composition of each tissue reported in the original paper, but also enabled the identification of tissue-specific features, even though the reference only contained splenic samples.

While there were no other available methods to perform “projection” of query data into a reference atlas at the time of our submission (that is, query projection that does not alter the reference space), a recent pre-print (Oct 2020, <https://www.biorxiv.org/content/10.1101/2020.10.12.335331v1>) from the Satija lab describes a projection algorithm (“Azimuth”) similar to the one implemented in ProjectTILs. We have included a comparative analysis of Azimuth and ProjectTILs, first by using the default reference atlas from Azimuth (human PBMC, new Supplementary Fig. 12), as well as adapting it to run with our own reference TIL atlas (new Supplementary Fig. 13). We note that Azimuth is limited to the projection task, and does not include the additional layers of analyses for detection of novel/altered states that are offered by ProjectTILs.

Finally, we would also like to stress that the relevance of our work is not limited to the development of the projection algorithm; but rather to proposing a novel framework of single-cell data analysis where i) multi-study reference atlases are generated for a given cell type (we have focused on T cells); and ii) new data can be interpreted in the context of a stable reference atlas, both in terms of pre-annotated subtypes (equivalent to “mapping and label transfer”) and in terms of unique transcriptional alterations in the high-dimensional gene expression space (i.e. similar to the ‘unknown or rare cell-type identification’ problem).

Furthermore, in this work we have constructed two multi-study expert-curated T cells reference atlases for cancer and viral infection, which represent a fundamental aspect of our work. These atlases are by themselves valuable resources for the immunology community, and their associated Web apps (<http://tilatlas.unil.ch/> and <http://virustcellatlas.unil.ch/>) are actively used by our experimental collaborators and several immunology labs. As demonstrated in our manuscript, high-resolution T cell atlases allowed ProjectTILs to recapitulate the observations from several recent studies covering a wide range of experimental settings and perturbations in mouse and human (multiple gene knock-outs, CD4 depletion, antigen-specificity, tissue composition, etc.).

In the revised manuscript, we discuss and cite other methods in the introduction, and attempted to better convey how our proposed approach differs from available tools.

From line 61:

A second outstanding challenge in single-cell data science is the mapping of single cells to a reference atlas¹⁶, and several methods have recently been proposed to address this task^{17–20}. These methods allow mapping cluster annotations of a reference atlas to individual cells of a query dataset (also referred to as “label transfer”), defining a new embedding space specific for the query. A recently released algorithm from the Seurat developers (Azimuth²¹) allows preserving the integrity of the

reference atlas space upon projection, and query datasets can be interpreted in the context of such reference atlas.

From line 74:

In contrast to other methods, ProjecTILs allows not only projecting new data into a reference atlas without altering the reference space, but also detecting and characterizing previously unknown cell states that “deviate” from the reference subtypes.

2. In the benchmark part, the authors only show the robustness of the method by cross-validation experiments. The authors should compare with the other methods, including but not limited to scmap.

Response - Thank you for the suggestion. We have now included a predictive performance comparison to Azimuth (a pipeline recently released by the Seurat authors for dataset projection, see also previous response) and to scMap for the cross-validation experiments. We found that all methods do a reasonable job for classifying T cells previously removed from the reference, but that ProjecTILs significantly outperforms Azimuth and scMap (Supplementary Fig. 3).

Moreover, we evaluated Azimuth for the projection and classification of human T cell data on the patient cohort from Yost et al. We applied Azimuth both on the PBMC reference atlas provided by the authors, as well as adapting the method to use our own TIL reference atlas. While broad T cell states (CD4 vs. CD8, Tregs) could be distinguished on the PBMC reference atlas, Azimuth could not discern between more specific subtypes. A considerable fraction of cells were also assigned to the NK and MAIT areas of the reference map (Supplementary Fig. 12).

We then applied Azimuth to project the Yost et al. data onto the TIL reference atlas constructed by us. Projection was reasonable for some T cell subtypes (Treg, Tfh, CD8 effector), but naive-like cells were mostly misclassified, as well as exhausted T cells (Supplementary Fig. 13 A-B). On a subset of cell subtypes that could be confidently mapped between studies, Azimuth resulted less accurate than ProjecTILs for the classification task (Supplementary Fig. 13 A-B). These results highlight the importance of an accurate projection algorithm, but also of using a robust atlas specific for the problem at hand, rather than generalist, whole tissue atlases.

Finally, we would like to stress that the robustness of our method has been demonstrated in multiple scenarios throughout our paper. First, by validating the results of multiple previous studies (e.g. miR155-KO, Regnase1-KO), and also with the new analysis of tissue-specific T cell phenotypes in chronic infection (new Figure 5); then by comparing T cell classification by ProjecTILs projection with the original subtype annotation from human studies (e.g. Yost, Li). We applied ProjecTILs both to a TIL reference atlas in cancer, as well as to a CD8 T cell atlas for chronic and acute infection. We have included several new analyses in the updated manuscript. The consistency of classification across studies, tumor models and even species (new Figure 7) attests to the robustness of the method in a range of different scenarios.

3. scRNA-seq data may contains unknown cell types that are not reported by the references. How is the performance of ProjecTILs in this situation? The authors should also compare the results with the other software, such as MARS.

Response - To the best of our knowledge, ProjecTILs is the first algorithm for projection of new data into a stable reference atlas that preserves the integrity of the reference space and also allows to detect altered cell states that deviate from the reference – in other words, unknown or rare subtypes. Please note that we refer to “subtypes” or “cell states” rather than “cell types”, because our atlases are cell-type specific (T cells in this case).

Other methods, such as MARS (Brbic et al. Nature Methods 2020), can learn cell type features from one or several labelled reference datasets, and then automatically annotate cell types in another (query) dataset, including novel cell types. These can then be interpreted in a new, query-dependent embedding space. However, MARS does not enable projection of the query dataset in the stable, unaltered transcriptional space of the reference, which is an essential premise for ProjecTILs. In the updated manuscript, we have added performance comparisons (for the projection and classification task) with Azimuth and scMap (see Supplementary Fig. 3, 12 and 13), and mentioned MARS and other tools in the updated introduction.

We have shown several examples where ProjectTILs could be applied to interpret altered cell states, including miR-155 KO, Regnase KO, Ptpn2 KO, CD4 depletion, TOX KO. In the revised manuscript, we have also included additional examples that demonstrate the ability of ProjectTILs to detect cell states that deviate from the reference, including the human study suggested by reviewer #2 (Auxiliary Figure 2) and the recent study by Sandu et al. (Figure 5).

4. In Figure 2, the authors project a scRNA-seq dataset onto the reference TIL atlas, a dataset of tumor-infiltrating T cells. The authors only analyze the component of subclones in each cluster without explaining more detailed results about the association between clusters and subclones. The authors may find another time-course dataset to show the relationships. Furthermore, Figure 2G is unclear, such as the definition of the level of clonal expansion.

Response - Figure 2 has the main purpose of providing a schematic representation of the algorithm, showcasing the functionalities of ProjectTILs using a representative example. We realize that the example in panels D to G may have been unclear, and have replaced it (as also suggested by reviewer #1) with the analysis of tetramer+ CD8 TILs from Miller et al.

As for the relation between reference clusters and T cell clones, we investigated this point in detail using human T cell data in the section “*Clonotype meta-analysis suggests that human tumor-specific EM CD8 T cells differentiate into exhausted/dysfunctional TILs through a progenitor subtype that upregulates TOX*”, where we analysed single-cell TCR clonal overlap between TIL states in two patient cohorts. We found a strong overlap between Tex and T_{pex} repertoires in both cohorts (Figure 8 D-E), consistent with the notion from mouse studies that T_{pex} cells give rise to Tex cells. Moreover, we observed clonal relatedness also between the (precursor) exhausted and effector-memory subtypes, prompting us to suggest a potential differentiation path that connects these three subtypes.

5. ProjectTILs uses PCA or ICA for dimension reduction. Please set some criteria to evaluate the performance in different data scenarios.

Response - As detailed in the updated methods, we always use PCA for dataset projection and calculation of the UMAP embeddings. ICA components are only used to evaluate deviations from the reference states (in what we termed “discriminant component analysis”). This choice is motivated by the fact the ICA detects independent modules that we can loosely associate with individual genetic programs (see the ICA signature enrichment analysis in Supplementary Fig. 4). Furthermore, because the main axes of variation of the UMAP are determined over a PCA dimensionality reduction, we reason that ICA provides a complementary source of information for the detection of altered gene programs, compared to PCA.

Updated Methods’ description, from line 627:

Because ProjectTILs relies on PCA for dimensionality reduction and projection, we reasoned that ICA components could provide a complementary and non-redundant decomposition of transcriptomics signals compared to PCA. We applied the fastICA implementation to determine 50 independent components on the integrated expression matrix of the TIL reference atlas. To suggest a biological interpretation of the ICA components, we downloaded hallmark gene sets (H) and canonical pathway gene sets (CP) from the Molecular Signatures Database (mSigDB), as well as selected immunological signatures from previous studies. We scored each ICA against these signatures by summing the ICA gene loadings for all genes in a given signature, and then taking the absolute value of this score. We retained the top-three scoring signatures for each ICA, and clustered ICA components based on the union of all retained signatures (see Supplementary Fig. 4).

6. Please use the consistent wording in the paper: for example, CD8 EM, CD8_EM, CD8_EffectorMemory may refer the same cluster.

Response - Thank you for pointing this out; we have updated the nomenclature to be consistent throughout the text, or explicitly specified alternative names when necessary.

REVIEWERS' COMMENTS

Reviewer #1 (Remarks to the Author):

All of our concerns were addressed successfully and we have no further comments.

Reviewer #2 (Remarks to the Author):

The authors have sufficiently addressed my concerns.

Reviewer #3 (Remarks to the Author):

The authors have addressed my concerns.